# HAND1 level controls the specification of multipotent cardiac and extraembryonic progenitors from human pluripotent stem cells

Adam T Lynch[1,4], Naomi Phillips [ID] [1,4], Megan Douglas [ID] [1], Marta Dorgnach[1], I-Hsuan Lin[1], Antony D Adamson [ID] [1], Zoulfia Darieva[1], Jessica Whittle[1], Neil A Hanley [ID] [1,2,3], Nicoletta Bobola [ID] [1] & Matthew J Birket [ID] [1✉]

## Abstract

**Diverse sets of progenitors contribute to the development of the embryonic heart, but the mechanisms of their specification have remained elusive. Here, using a human pluripotent stem cell (hPSC) model, we deciphered cardiac and non-cardiac lineage trajectories in differentiation and identified transcription factors underpinning cell specification, identity and function. We discovered a concentration-dependent, fate determining function for the basic helix-loop-helix transcription factor HAND1 in mesodermal progenitors and uncovered its gene regulatory network. At low expression level, HAND1 directs differentiation towards multipotent juxta-cardiac field progenitors able to make cardiomyocytes and epicardial cells, whereas at high level it promotes the development of extraembryonic mesoderm. Importantly, HAND1-low progenitors can be propagated in their multipotent state. This detailed mechanistic insight into human development has the potential to accelerate the delivery of effective disease modelling, including for congenital heart disease, and cell therapy-based regenerative medicine.**

**Keywords** HAND1; Epicardial; Single Cell; Gene Regulatory Networks; Juxta-cardiac Field
**Subject Categories** Chromatin, Transcription & Genomics; Development

## Introduction

Development of the mammalian four-chamber heart is highly complex and depends on the specification and exquisitely regulated, phased differentiation of multiple populations of progenitor cells to a broad final array of muscular (cardiomyocytes) and non-muscular lineages (Meilhac and Buckingham, 2018).

Errors in any part of this process have been catalogued in human congenital heart disease and have illustrated necessary aspects of genetic regulation (Houyel and Meilhac, 2021; Morton et al, 2022). However, how a number of the critical early human embryonic fate decisions are made has remained elusive.

During gastrulation, cell diversity along the primitive streak is largely enabled by differential intensity and duration of the two main branches of the TGF-β signalling pathway mediated by BMP and Nodal ligands, and activating specific R-SMADs (Bardot and Hadjantonakis, 2020; Jia and Meng, 2021). Cardiogenic mesodermal progenitors arise under the combined actions of BMP, Nodal/Activin, and WNT signalling and migrate from the primitive streak in two broad waves (Devine et al, 2014; Lescroart et al, 2014; Meilhac et al, 2004). The initial wave forms the first heart field (FHF) and leads to the linear heart tube (Ivanovitch et al, 2017). Subsequently, second heart field (SHF) progenitors contribute progressively to the heart's reshaping into its final four-chamber structure. Within this framework, transcription factors (TFs) controlling the identity of terminally differentiated cells such as cardiomyocytes have been well described. However, how an overlapping set of TFs specify progenitors, including those retaining multipotency, is poorly understood. Recently, single cell genomics, including directly in human embryos, has reinforced the idea that progenitors contributing to the FHF and SHF are actually made up of subpopulations with discrete identities and fate assignments (Dominguez et al, 2023; Ivanovitch et al, 2021; Tyser et al, 2021b), such as the juxta-cardiac field (JCF) which overlaps the FHF and is marked by the basic helix-loop-helix (bHLH) TF, Hand1. The JCF gives rise to cardiomyocytes and epicardial cells in the heart but also contributes cells to the extraembryonic mesoderm (Tyser et al, 2021a; Zhang et al, 2021). Hand1 is functionally important in each cell type in this array and global Hand1-null mice die by E8.5 due to a deficiency in extraembryonic mesoderm (Firulli et al, 1998, 2020; Riley et al, 1998; Risebro et al, 2006; Vincentz et al, 2017; Li et al, 2017). HAND TFs are considered members of the evolutionarily conserved core cardiac TF network alongside other ancient TFs including MESP1, GATA4,

[1]Faculty of Biology, Medicine and Health, University of Manchester, Manchester, UK. [2]College of Medicine & Health, University of Birmingham, Edgbaston, Birmingham B15 2TT, UK. [3]University Hospitals Birmingham NHS Foundation Trust, Birmingham B15 2GW, UK. [4]These authors contributed equally: Adam T Lynch, Naomi Phillips.
✉E-mail: matthew.birket@manchester.ac.uk

MEF2C, TBX5 and NKX2-5 (Kathiriya et al, 2015). However, the underlying mechanisms that determine lineage commitment and control multipotency within progenitor cells remain unresolved, hampering effective regenerative medicine and disease modelling requiring the accurate programming of specific human cardiac cell types.

In this study, we aimed to address this knowledge gap by using human pluripotent stem cells (hPSCs) to model a wide repertoire of cardiac cell development and explore whether a multipotent embryonic progenitor state could be propagated in vitro. Specifically, by using lineage tracking and single cell genomics, we aimed to discover the gene regulatory networks (GRNs) that were operational in specifying the different cardiac cell fates, and mechanistically support advances in the field where 3D models have been used to promote the development of multiple heart cell types (Meier et al, 2023; Schmidt et al, 2023; Lee et al, 2017).

The outcome established an upstream role for HAND1 in programming the specification and GRNs of multipotent progenitors within the JCF and related extraembryonic mesoderm. The data supports a TF concentration-dependent mechanism of cell fate determination.

## Results

### Multi-lineage cell diversity can be achieved in 3D by modulating the duration of Activin and BMP signalling

We hypothesized that mimicking the patterning of the primitive streak would achieve diverse differentiation including endoderm and non-cardiac embryonic and extraembryonic mesoderm alongside cardiogenic mesoderm. We adapted our previously characterized 3D cardiomyocyte differentiation model in hPSCs (Birket et al, 2015) and modulated the duration of Activin and BMP signalling with small molecule inhibitors (Fig. 1A). In the continued presence of exogenous Activin A and BMP4, the ALK5 inhibitor SB431542 (SB) and the ALK1-3 inhibitor DMH1 terminated receptor-regulated SMAD2/3 and SMAD1/5/8 phosphorylation, respectively (Fig. 1B). R-SMADs compete for common partner SMAD4, therefore we hypothesized these inhibitions would enhance the downstream activity of the uninhibited pathway. In line with this, SB promoted development of posterior primitive streak-like mesoderm with increased *TBXT*, *MESP1*, and *HOXB1 mRNA* expression compared to the control, whereas DMH1 suppressed *MESP1* and promoted anterior primitive streak-like mesoderm and endoderm development with increased *MIXL1* and *FOXA2 mRNA* (Fig. 1C). Immunostaining of FOXA2 and SOX17 confirmed that DMH1 treatment induced robust endoderm differentiation (Appendix Fig. S1A). To deconvolute this developmental diversity, we created a dual cardiac-endoderm fluorescent reporter hESC line by taking an *NKX2-5-GFP* reporter hESC line (Elliott et al, 2011) and introducing a knock-in *SOX17-T2A-dTomato* by CRISPR-Cas9 gene editing (Appendix Fig. S1B), thereby allowing the identification by flow cytometry of early cells in each lineage. With DMH1, SOX17-Tom expression became prominent at day 4 of differentiation and persisted over several days (Fig. 1D–G). By day 8, NKX2-5-GFP was also detectable with DMH1 whereas it was abolished by a saturating dose of SB, demonstrating a block to cardiac differentiation. To explore and understand the boundaries between

cardiac and other lineages, we first titrated the small molecules and measured the reporters in concert with surface markers PDGFRα (broad mesoderm) and CD235 (anterior-like mesoderm) (Fig. 1E) (Lee et al, 2017). At day 4, PDGFRα and CD235 showed an inverse pattern through the signalling gradient (Fig. 1E,F). At day 7.5, SOX17-Tom could be detected simultaneously with NKX2-5-GFP (Fig. 1E,G) and while mostly mutually exclusive, a population of Tom+ GFP+ cells was also detected (Fig. 1E). To help clarify the identity of this dual positive population we introduced a knock-in *FOXA2-T2A-mTagBFP2* reporter as a second endoderm marker (Appendix Fig. S1C). In this triple reporter, FOXA2-BFP was entirely co-expressed with SOX17-Tom at day 7.5 when NKX2-5-GFP+ cells emerged (Appendix Fig. S1D), consistent with data in early human embryos (Tyser et al, 2021b). However, *FOXA2* expression has also been reported in mouse cardiac mesoderm, thus still leaving some ambiguity as to the origin and identity of these cells (Ivanovitch et al, 2021; Bardot et al, 2017). To test if they included cardiac progenitors, we isolated SOX17 + FOXA2+ cells by FACS at day 5 for direct differentiation but found they generated very few cardiomyocytes suggesting these were largely not mesodermal but instead endodermal cells (Appendix Fig. S1E,F).

Establishing anterior-posterior primitive streak-like patterning allowed us to select cardiac-permissive concentrations of SB and/or DMH1 (at 0.6 μM and 0.06 μM, respectively). At these concentrations we noted that WT1+ mesothelial/epicardial cells developed in control conditions and following the addition of SB, but not following DMH1 treatment, whereas cardiomyocytes formed in all conditions (Fig. 1H). This led us to question what molecular mechanisms limit cardiac differentiation under suboptimal (high BMP or high Activin signalling) conditions. And prompted us to ask: what markers discriminate cardiac progenitors with different developmental potentials, including those with or without the ability to make epicardial cells?

Based on NKX2-5-GFP (G) and SOX17-Tom (T) expression at day 7.5, a timepoint when cardiac progenitors may still not be fully differentiated, we sorted 8 populations by FACS for RNA-seq which we hypothesized would be developmentally distinct and thus valuable to characterize, including cardiac and closely related lineages (Fig. 1E,I). Developmental marker analysis and hierarchical clustering showed that the G+ cells had a uniformly myocardial identity except when emerging from the SOX17+ lineage (G + T +), which had an endoderm signature specific of thymic epithelial cells (Fig. 1J) (Wei and Condie, 2011). Many 'cardiac progenitor' TFs including *GATA4/5/6*, *TBX5*, and *MEIS1/2* were expressed at similar levels across control and SB-treated cells independently of GFP expression. However, whereas G + T- cells in these conditions had a myocardial-commitment signature of high *SRF*, *MEF2C*, and *MYOCD* expression, the G-T- cells had higher *HAND1* and other markers described to demarcate the JCF, proepicardium, septum transversum mesenchyme, and extraembryonic mesoderm (Tyser et al, 2021a; Zhang et al, 2021). In contrast, with DMH1, emerging G + T- cells displayed an anterior SHF-like profile (Meilhac and Buckingham, 2018) with high *FGF8* and *FGF10* expression and an absence of posterior *HOXB1*. Expression values for some key markers are shown in Appendix Fig. S1G. In summary, these data confirm effective anterior-posterior primitive streak-like patterning and the programming of different sets of cardiac progenitors alongside closely related cells.

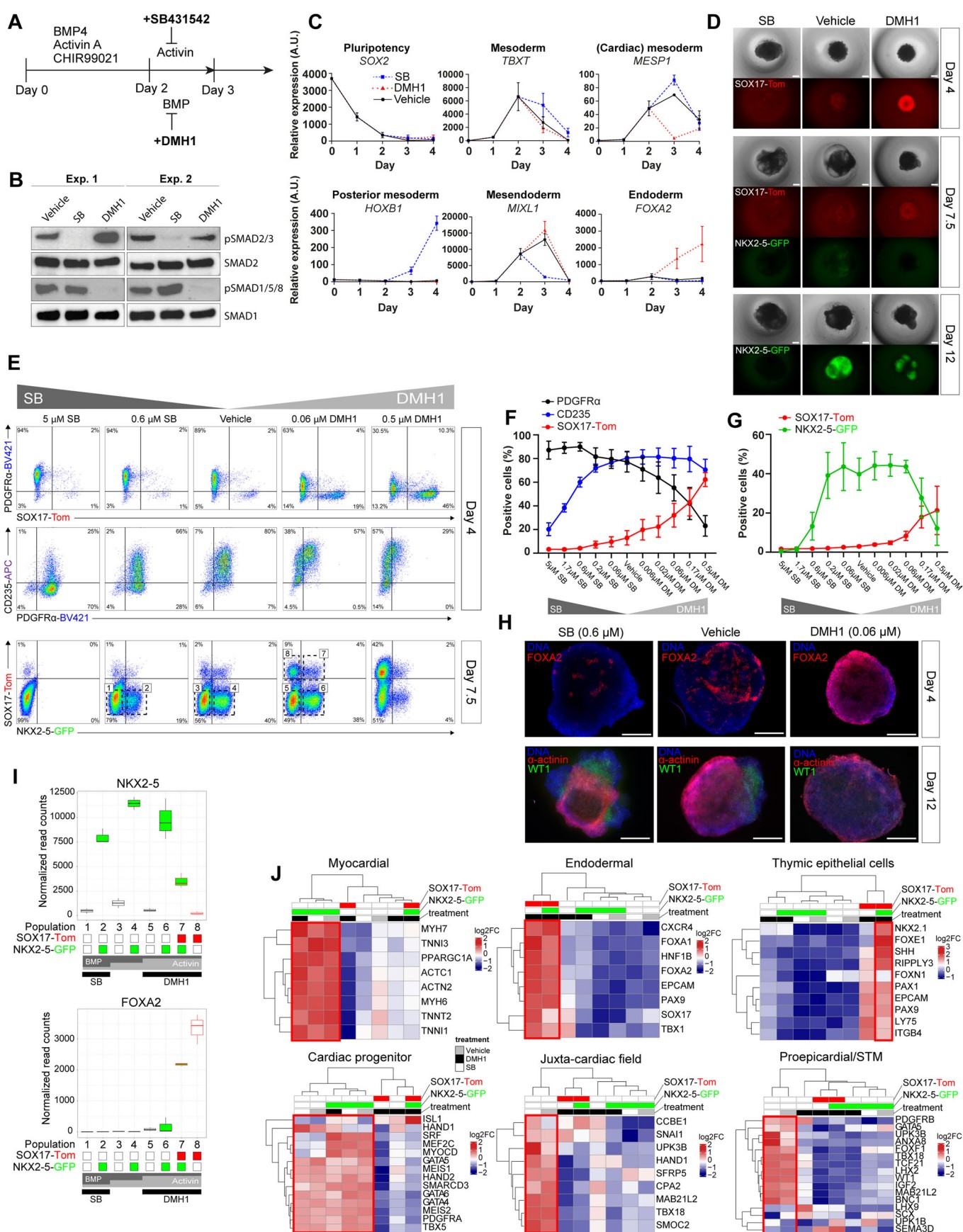

**Figure 1.   Multi-lineage cell diversity in 3D can be achieved by modulating the duration of Activin and BMP signalling.**

(A) Schematic of hESC 3D EB differentiation protocol with pathway inhibitors introduced at day 2 to promote cellular diversity. All factors were removed at day 3. (B) Western blots of receptor-SMADs 6 h after small molecule addition following the protocol in (A). (C) RT-qPCR analysis of marker gene expression in a differentiation time-course following the protocol in (A). (D) Live imaging of SOX17-Tom and NKX2-5-GFP at 3 timepoints of differentiation. (E) Representative flow cytometric analyses of surface markers PDGFRα and CD235, together with SOX17-Tom and NKX2-5-GFP at 2 timepoints of differentiation with the indicated concentrations of SB and DMH1. (F) Positive cell proportions for day 4 and (G) for day 7.5. (H) Immunostaining of whole mount EBs stained for FOXA2 (red) at day 4, and α-actinin (red) and WT1 (green) at day 12. (I) RNA-seq quantifications of *NKX2-5* and *FOXA2* after the sorting of 8 populations at day 7.5 based on SOX17-Tom and NKX2-5-GFP. The boxplots follow standard Tukey representations ($n = 3$ independent biological experiments) and are coloured by the lineage markers. The top and bottom edges of the boxplots represent the upper (75th percentile) and lower (25th percentile) quartiles. Horizontal lines within each box represent the median. Whiskers extend to 1.5 times the interquartile range. Typical population sorting gates are indicated in (E). (J) Hierarchical clustering of cell type and lineage markers based on RNA-seq data from the 8 populations. Expression levels are represented as Z-score normalized log2 FC differences. Data are represented as mean ± SEM ($n = 3$–4) independent biological experiments for (C, F, G). Scale bars represent 300 µm. A.U. arbitrary units, EB embryoid body, Exp. experiment, FC fold change, SEM standard error of the mean, STM septum transversum mesenchyme. See also Appendix Fig. S1. Source data are available online for this figure.

## HAND1 and HOX transcription factors regulate the cardiac inhibitory effect of high BMP signalling

We next performed chromatin accessibility analysis on the isolated populations to gain insight into the transcription factors regulating lineage identity (Appendix Fig. S2A, B). To obtain epigenetic signatures, we collated all regions which were unique to broad population classes based solely on NKX2-5-GFP and SOX17-Tom status, i.e. G-T-, G + T-, G-T+ and G + T+, as detailed in Appendix Fig. S2C. Motif enrichment analysis was performed (Fig. 2A). Binding motifs of GATA, CTCF, MEIS/PBX, MEF2, NKX2-5, and TBX20 were significantly enriched in the cardiac G + T- population. In contrast, motifs of TEAD, WT1, HOX, and HAND were enriched in the G-T- population. This was consistent with the expression of these TFs and their local chromatin state (Appendix Fig. S2D–H). We attribute the HAND motif enrichment to the expression of *HAND1*, and the HOX motif enrichment primarily to the expression of the HOXB cluster, principally *HOXB1–6*, those being the predominantly expressed family members in that population class (Appendix Fig. S2D–F). The WT1 motif enrichment may be explained by the corresponding presence of epicardial cells, but there is also evidence for WT1 playing a role in cardiac progenitors (Marques et al, 2022; Martínez-Estrada et al, 2010). The G-T+ population was dominated by the enrichment of motifs and associated co-expression of endoderm TFs FOXA2, SOX17, and HNF4A (Fig. 2A; Appendix Fig. S2D, G). The G + T+ population additionally showed a significant enrichment of the motif for NKX2-5, confirming its functional role in these cells (Fig. 2A).

Considering these data, we hypothesized that some of the TFs active in the G-T- mesodermal cells in the high BMP signalling environment, i.e. HAND1, TEAD, HOX, and WT1, might have an important role in the specification of JCF or extraembryonic mesodermal progenitors, and so potentially determine cell fate in this high BMP environment and direct cells away from becoming cardiomyocytes (NKX2-5-GFP +). To test this, we made a series of gene knockout hESC lines including *YAP1*, *WT1*, *HOXB1–3* and *HAND1* by CRISPR-Cas9 gene editing (Appendix Fig. S3A–F). To overcome the genetic redundancy of TEAD and HOX, which have 4 and 39 family members, respectively, we created YAP1-null hESCs to inhibit the formation of TEAD-YAP1 complexes, and generated a triple knockout of *HOXB1-2-3* (HOXB1–3-null) to restrict HOX activity (Appendix Fig. S3A–C). The impact of each TF type on cardiac differentiation and NKX2-5-GFP expression

was assessed (Fig. 2B–E). Although YAP1-null cells could express the mesoderm marker PDGFRα at day 4 (Appendix Fig. S3D), they then ceased to proliferate and failed to express any GFP even under control conditions. In contrast, the WT1-null, HOXB1–3-null and HAND1-null hESCs could all differentiate to cardiomyocytes under control conditions, with GFP levels similar to the wild-type (Fig. 2B,C). The WT1-null behaved like the wild-type in terms of repression of cardiac differentiation by SB, whereas the HOXB1–3-null and the HAND1-null showed significant resistance to SB, which in the case of the HAND1-null was essentially complete (Fig. 2D). WT1+ epicardial cells were completely absent in the HAND1-null where the EBs instead contained an increased number of α-actinin+ and CTNT+ cardiomyocytes, consistent with the increased NKX2-5-GFP expression (Fig. 2E; Appendix Fig. S3G). We confirmed the absence of WT1+ epicardial cells in an additional HAND1-null clone of the same parental line, demonstrating the HAND1-null phenotype was not clone-specific (Appendix Fig. S3H). To check the importance of genetic background, we made a *HAND1* knockout in an independent hESC line carrying a 5' *GFP-T2A-NKX2-5* knock-in reporter (Appendix Fig. S3I) (Lynch et al, 2024). From a poor baseline differentiation efficiency to GFP+ cells in 3D format, we found that the *HAND1* knockout in this line dramatically improved GFP+ cardiomyocyte differentiation efficiency and abolished WT1+ cells (Appendix Fig. S3J,K), supporting the universality of this mechanism.

These data position HAND1 and the HOXB family as fate determining factors in a high BMP signalling environment and suggest that HAND1 may play a major role in the development of JCF-like multipotent progenitors. At higher levels, HAND1 and the HOXB family may block cardiac fate and potentially drive an extraembryonic fate.

## Single-cell RNA-sequencing supports a functional role for HAND1 in the development of a multipotent 'JCF'-like population

To gain insight into the role of HAND1 in cardiac and non-cardiac progenitors, we characterized the cellular identities in the wild-type and HAND1-null cell models during differentiation by single-cell RNA-sequencing. To promote the development of a broad range of cell types, we combined populations of vehicle, SB-treated and DMH1-treated cells (as described above) at a ratio of 0.5:0.25:0.25, respectively, and collected cells at 7 timepoints. Using UMAP with

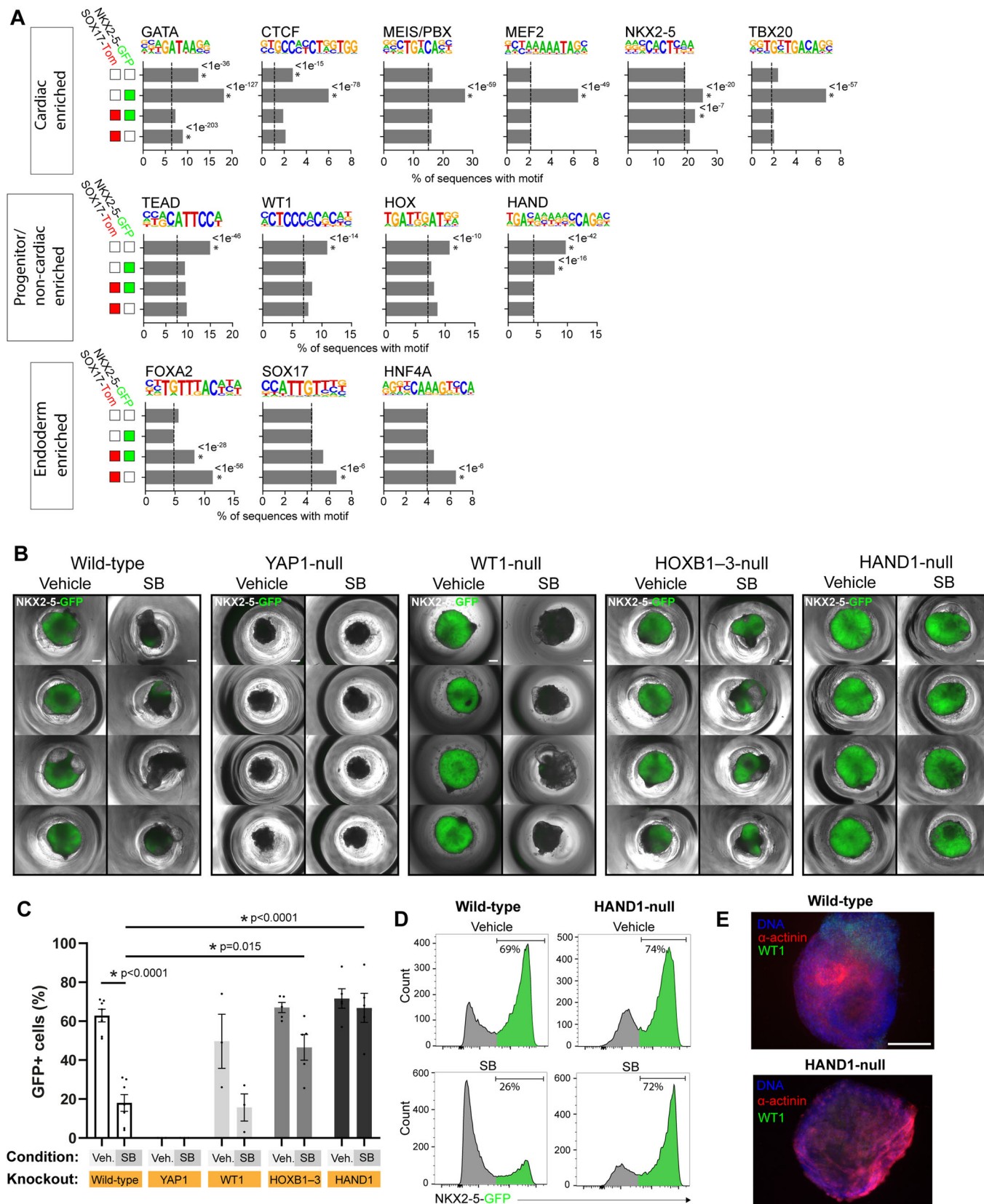

**Figure 2.    HAND1 and HOX transcription factors regulate the cardiac inhibitory effect of high BMP signalling.**

(A) Transcription factor motif enrichment analysis of ATAC-seq data from sorted cells at day 7.5. The results are from an input of all regions of open chromatin uniquely identified in each of the 4 population classes as indicated by the coloured boxes and based only on SOX17-Tom and NKX2-5-GFP status. For each motif, the average background value is indicated by the dotted line. Significance above background for any given population using a binomial test *$p < 1e-6$ (exact $p$ values as indicated). (B) Typical EBs at day 10 under control or SB-treated (day 2–3) conditions, from wild-type, YAP1-null, WT1-null, HOXB1–3-null, and HAND1-null hESCs. The brightfield image is overlayed by the NKX2-5-GFP signal, with quantification by flow cytometry shown in (C). Each point displays the result of an independent biological experiment. Data are represented as mean ± SEM ($n = 3$–7 independent biological experiments). Significance between any population was assessed by One-Way ANOVA with Tukey's multiple comparison test *$p < 0.05$. (D) Typical flow cytometric analysis of GFP fluorescence in wild-type and HAND1-null EBs at day 14 with and without SB treatment. (E) Immunostaining of whole mount day 12 EBs stained for α-actinin (red) and WT1 (green). Scale bars represent 750 μm in (B), 300 μm in (E). EB embryoid body. See also Appendix Figs. S2, S3. Source data are available online for this figure.

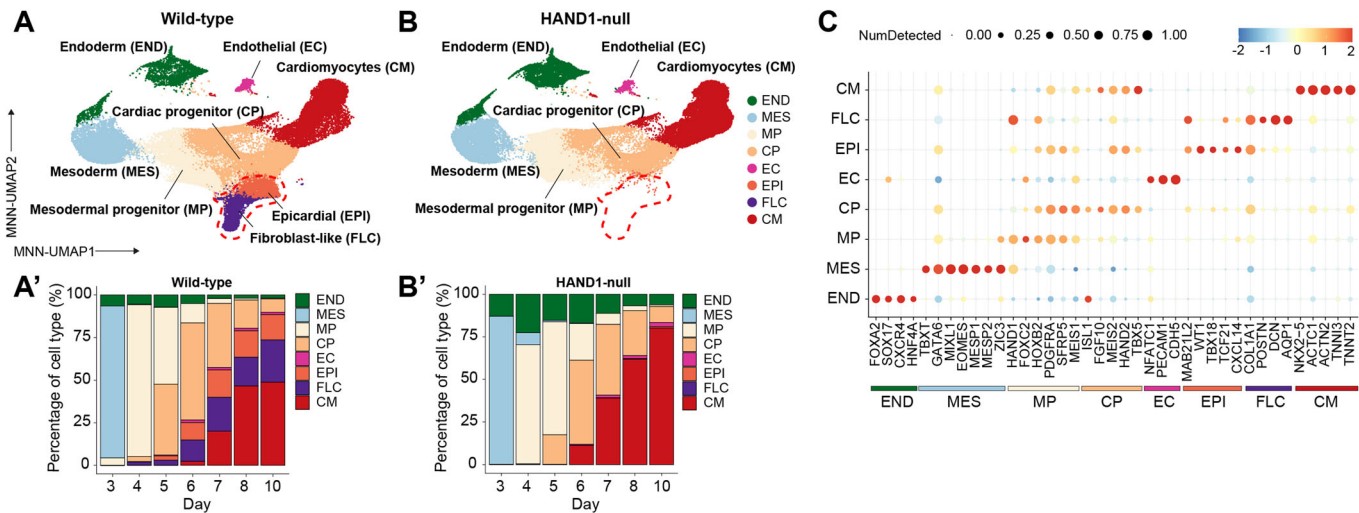

**Figure 3.    Single-cell RNA-sequencing shows that HAND1 is important for epicardial and fibroblast-like cell development.**

MNN-UMAP plots of (A), wild-type and (B), HAND1- null cells from day 3–10 of differentiation coloured by cell type. The dashed red lasso highlights the loss of epicardial and fibroblast-like cells in the HAND1-null. (A') The frequency of each cell type by day in wild-type and (B') HAND1-null. (C) The levels of prominent markers identifying each population in wild-type cells.

mutual nearest neighbour (MNN) correction, unsupervised clustering and detailed marker analysis on the combined transcriptomes of >100k wild-type and HAND1-null cells, we could annotate 8 broad population types: endoderm (END), mesoderm (MES), mesodermal progenitors (MP), cardiac progenitors (CP), endothelial (EC), cardiomyocytes (CM), epicardial (EPI), and fibroblast-like cells (FLC) and calculated assignment frequencies by day for wild-type (Fig. 3A,A') and for HAND1-null (Fig. 3B,B'). Figure 3C shows the levels of prominent markers of each cell population in the wild-type. Whereas the populations in the upper part of the MNN-UMAP plots are largely overlapping (Fig. 3A,B), there was a clear absence of epicardial and fibroblast-like cell development in the HAND1-null. Thus, HAND1 promotes epicardial and fibroblast differentiation, consistent with the previous data (Fig. 2E).

To gain further biological insight into cardiac cell lineage development, we performed line-specific analyses using only hyper variable genes (HVGs) implicated in cardiac development (~1000 genes), while retaining the previous cell type annotations. Monocle 3 was used to make new UMAP plots for each line, and to assign pseudotime and lineage trajectory data from root mesoderm (Trapnell et al, 2014). In the wild-type, the major mesoderm derivatives still partitioned into four terminal lineages (Fig. 4A). Lineage trajectory analysis enabled the inference of their

developmental origins (Fig. 4A'). From differential gene expression analysis of the two lineages, we deduced that the early bifurcation point in the Monocle trajectory may represent an FHF- and SHF-like split in the mesoderm (Appendix Fig. S4A,A'). We noted *HAND1*, *TBX2* and *HCN4* expression on the FHF-like lineage (Fig. 4B). *TBX2* lineage tracing showed it marks cells which make a similar contribution to the left ventricle as *HAND1* (Aanhaanen et al, 2009), and *HCN4* is recognized as an FHF progenitor marker (Liang et al, 2013). The SHF-like lineage showed prominent *ISL1*, *FGF10*, *FOXC1/2* and *JAG1* expression (Fig. 4B), which are well-established SHF markers (Cai et al, 2003; Dominguez et al, 2023; High et al, 2009; Kelly et al, 2001). Whereas the SHF-like lineage appeared to progress solely to cardiomyocytes, the FHF-like lineage —which at its early stage included a strong JCF-like signature (*HAND1*, *MAB21L2*, *BNC2* and *MSX1/2*) (Dominguez et al, 2023; Tyser et al, 2021a; Zhang et al, 2021)—displayed multilineage fate also giving rise to epicardial and fibroblast-like cells. Additionally, we detected a side branch from the FHF/JCF-like progenitors with high transient *CDX2*, *GATA2*, and *TBX3* expression and low *TBX5*, which may represent an extraembryonic mesoderm-like lineage (Fig. 4A',B) (Bulger et al, 2024).

In contrast to the wild-type, the HAND1-null showed a single dominant trajectory to cardiomyocytes (Fig. 4C,C',D). A small

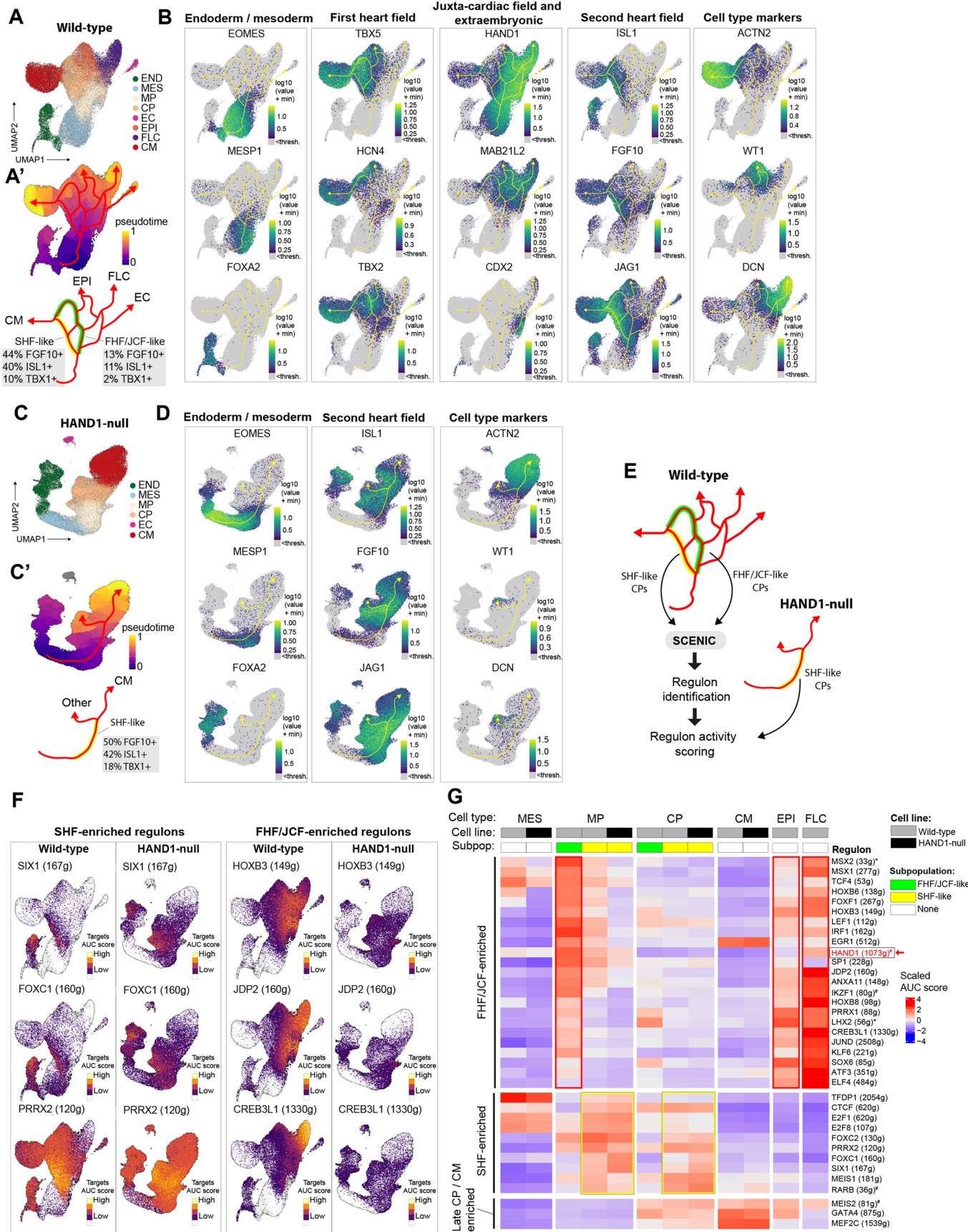

**Figure 4. Single-cell RNA-sequencing supports a functional role for HAND1 in the development of a multipotent 'JCF'-like population.**

(A–D) UMAP plots from Monocle3: (A) coloured by cell type in the wild-type and (C), in the HAND1-null line; (A') coloured by pseudotime with predicted trajectories overlayed, in the wild-type and (C') in the HAND1-null line. Endpoint cell types and FHF/JCF- and SHF-like populations have been annotated and their marker frequencies shown. (B) Gene expression of lineage and cell type markers for the wild-type and (D) HAND1-null lines. (E) SCENIC GRN analyses of cardiac progenitors extracted from the two lineage trajectories of the wild-type differentiation and the single lineage of the HAND1-null trajectory. (F) Activity of selected regulons enriched in cardiac progenitor lineages mapped to UMAP plots of wild-type and HAND1-null cells. White indicates the regulon activity is below threshold. The number of gene targets in each regulon is indicated following the name of the TF. (G) Heatmap of relative regulon activity by cell line, cell type, and subpopulation, grouped by lineage and stage of enrichment. The cardiac progenitor data represent cells extracted from each lineage. *Identified in HAND1-null population. #Identified in CPC lineage analysis. DEG differentially expressed genes, FHF first heart field, JCF juxta-cardiac field, SHF second heart field, GRN gene regulatory network, SCENIC single-cell regulatory network interference and clustering. See also Appendix Fig. S4.

number of WT1+ (EPI) and DCN+ (FLC) cells clustered together, but the development of FHF/JCF-like and extraembryonic-like progenitors and fibroblast-like cell derivatives, was almost entirely lost. The main lineage of progenitors in the HAND1-null had a SHF-like signature with an equivalent or higher frequency of *FGF10+* (50% vs 44%) *ISL1+* (42% vs 40%), and *TBX1+* (18% vs 10%) cells compared to the wild-type SHF-like lineage (Fig. 4A',C'). We confirmed the reproducibility of this change in cell fate from FHF/JCF to SHF by demonstrating a consistent increase in SHF markers *TBX1, JAG1* and *ISL1*, and a decrease in the FHF marker *TBX5* in day 5 HAND1-null progenitors in three further independent differentiations (Appendix Fig. S4B).

Following the assumption that the main differences between the wild-type and HAND1-null differentiations can be explained by a change in cardiac lineage selection and, therefore, by the gene regulatory networks (GRNs) coordinating these lineages, we performed GRN analysis using SCENIC to identify active TFs and their target genes (regulons) (Fig. 4E) (Aibar et al, 2017; Van de Sande et al, 2020). To maximize regulon detection power but also ensure robustness, we performed analyses on (1) all wild-type cells, (2) wild-type CP lineages-only (Appendix Fig. S4a), and (3) all HAND1-null cells, and ran the algorithm multiple times retaining only regulons identified in every run. A total of 307 regulons met these criteria (Fig. EV1 and Dataset EV1). Using our lineage analysis, we first focused on regulons with FHF/JCF- or SHF-bias in activity and examined these in the wild-type and HAND1-null populations (Fig. 4F,G). The SHF-like lineage was characterized by increased activity of FOXC1/2, SIX1, PRRX2 and MEIS1 regulons, and regulons controlled by regulators of cell proliferation including E2F family members (Fig. 4F,G). In contrast, the early FHF/JCF-like lineage was characterized by increased activity of HOX, JDP2, MSX1/2, FOXF1 and LEF1 regulons and many other regulons which were later enriched in the epicardial or fibroblast-like cells such as CREB3L1 (Fig. 4F,G). The HAND1-null progenitor lineage most resembled the SHF-like lineage of the wild-type and had lower activity of the FHF/JCF-associated regulons (Fig. 4F,G). A HAND1 regulon was identified in the wild-type cardiac progenitor analysis (Fig. 4G) but not in analyses of the full datasets where the regulon was lost due to lack of motif enrichment detected using cisTarget (Aibar et al, 2017). MESP1, another bHLH TF with an established role in cardiac mesoderm, was similarly lost, suggesting an insensitivity of the method to correctly prune/retain the regulons of some bHLH TFs using the standard gene regulatory space.

Our SCENIC analysis allowed us to rank the most specific regulons for each cell type (Fig. EV2). Comparing the cardiomyocytes in the wild-type to the HAND1-null revealed that the

dominant regulons were controlled largely by the same factors; this included classical cardiac TFs MEF2, SRF, GATA4, and TBX5; and metabolic regulators including estrogen-related receptors (ESRRA and ESRRG) and the transcriptional co-activator PGC-1α (PPARGC1A). These data support the conclusion that the major function of HAND1 in this system is not in determining the gross identity of cardiomyocytes but rather in controlling the type of mesodermal progenitors that are made.

## HAND1 controls chromatin and enhancer landscapes to pattern mesoderm and programme cell fate

To investigate how HAND1 functions in mesodermal progenitors, we first epitope-tagged endogenous *HAND1* in wild-type cells to map its DNA binding by ChIP-seq (Appendix Fig. S5A). HAND1 could be detected already by day 3 and was increased by SB treatment (Fig. 5A). At day 3, we identified 25,517 and 34,211 HAND1 binding sites in control and SB conditions, respectively, largely in distal intergenic and intronic locations (Appendix Fig. S5B). HAND1-bound regions were enriched for the previously identified long non-classical E-box HAND1 motif, but also for the recognition motifs of other factors known to be important in mesoderm including GATA6, EOMES, ZIC3 and TBX6 (Fig. 5B). Next, to investigate how HAND1 activity influences chromatin structure and gene expression we employed a tightly controlled system in which HAND1 could be transiently expressed at the mesoderm stage of HAND1-null cells. To do this we introduced a doxycycline (dox)-inducible *HAND1-BFP* transgene via lentivirus into ~30% of the cells. Dox was added at day 2.5 of differentiation, and 12 h later we isolated HAND1-BFP+ and BFP-negative cells for ATAC-seq and RNA-seq (Fig. 5C). HAND1 induced a dramatic restructuring of the chromatin, largely in distal intergenic and intronic locations, with ~17k regions displaying significantly increased accessibility and ~9k regions displaying significantly decreased accessibility (Fig. 5C'; Appendix Fig. S5B). A total of 1499 genes were significantly upregulated, and 1630 genes were significantly downregulated by the *HAND1* transgene (Dataset EV1). Integrating the ATAC-seq and ChIP-seq data revealed that HAND1 binding was enriched in the regions of increased accessibility (Fig. 5C'). Examining these genomic subsets allowed us to gain further insight into the TFs impacting the chromatin environment as a result of HAND1 expression. This showed that the above HAND1 motif was uniquely enriched in the regions of HAND1 binding with increased accessibility and not in the regions with decreased accessibility even where HAND1 was bound (Appendix Fig. S5C). HAND1 binding and chromatin opening was also specifically associated with the binding motifs of SMAD,

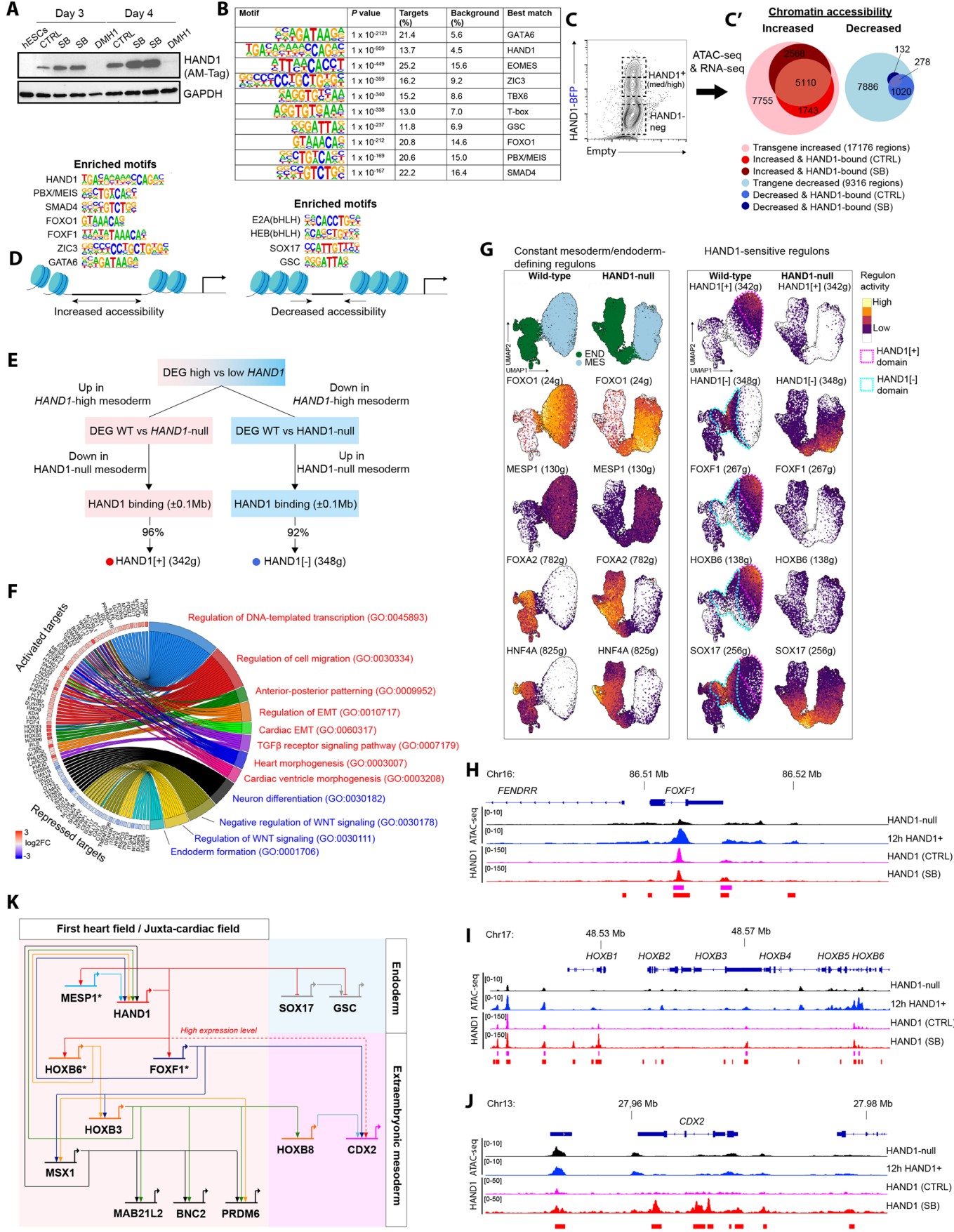

Figure 5. HAND1 controls chromatin and enhancer landscapes to pattern mesoderm and programme cell fate.

(A) Western blot of HAND1-AM-Tag (Active Motif) at day 3 and 4 of differentiation in different conditions compared to undifferentiated hESCs. (B) Motif enrichment analysis of HAND1 ChIP-seq peaks with control (low HAND1) and SB (high HAND1) sets merged. Significance is relative to the background control. (C) Doxycycline-inducible HAND1-BFP transgene expressed in HAND1-null cells for cell sorting by HAND1 expression (12-hour induction) and molecular analysis by ATAC- and RNA-seq (both performed on 3 biological replicates). (C') Euler plots for the ATAC-seq analysis representing the number of differentially accessible chromatin regions in HAND1+ vs HAND1-negative cells and the subset with detected HAND1 binding. (D) Schematic illustration of the motifs enriched in chromatin with increased or decreased accessibility as a result of HAND1 expression. (E) Identification of HAND1[+] (activating) and HAND1[-] (repressing) regulons. (F) Chord plot showing significantly enriched gene ontology terms for HAND1 target genes. The red and blue text colouring indicates terms enriched in activated targets and repressed targets, respectively. (G) UMAP plots showing the activity of constant and HAND1-sensitive regulons in endoderm and mesoderm of wild-type and HAND1-null cells. The activity of the HAND1[+] and HAND1[-] regulons are shown with the domains of strong activity highlighted. (H–J) ATAC-seq in HAND1-null and HAND1+ (12 h induction) populations, and HAND1 binding at low-HAND1 (CTRL) and high-HAND1 (SB) levels at the (H) FOXF1 locus, (I) HOXB cluster and (J) CDX2 locus. (K) Gene regulatory network model of HAND1. * Direct early target of HAND1. The dashed line indicates a link at high HAND1 levels. CTRL control/vehicle-only, DEG differentially expressed genes, EMT epithelial-mesenchymal transition, GO gene ontology, WT wild-type. See also Appendix Fig. S5. Source data are available online for this figure.

FOXO1, FOXF1, and PBX/MEIS. Conversely, regions of decreased accessibility displayed an exclusive enrichment of the classical E-box motifs CAGCTG and CACCTG, particularly in regions bound by HAND1. This suggests that the binding partners of HAND1 may determine its occupancy at different motifs. Regions of decreased accessibility were also enriched in motifs for GSC and SOX17, consistent with the downregulation of these genes (Appendix Fig. S5C). Supportive findings were observed in the analysis of published HAND1 ChIP-seq data from a later time-point in mesoderm differentiation (Tsankov et al, 2015). Figure 5D shows a schematic representation of these changes.

Next, we aimed to understand the HAND1 GRN in mesodermal progenitors. First, we identified high confidence target genes (regulons) using the strategy illustrated in Fig. 5E. The regulons comprised 342 activated targets and 348 repressed targets. 52% of the activated and 56% of the repressed genes were concordantly changed by the 12 h of *HAND1* transgene expression, suggesting those as probable early targets (Dataset EV1). Gene Ontology analysis provided an overview of the biological processes impacted by HAND1 (Fig. 5F). Activated targets were enriched for TFs and coactivators (including *MESP1/2*, *FOXF1* and *TEAD1*), genes involved in different aspects of heart morphogenesis, anterior-posterior pattern formation (including *HOXB2–9*) and regulators of cell migration and epithelial-mesenchymal transition (EMT) (including *SNAI1/2* and *CRB2*). Conversely, genes potentially repressed by HAND1 were associated with processes including neural development, WNT signalling and endoderm development (including *SOX17* and *NODAL*).

We used the gene-set analysis tool *AUCell* to quantify the activity of the activating HAND1[+] (342 g) and repressive HAND1[−] (348 g) regulons in our single cell data and compared the results to other relevant regulons (Fig. 5G). We noted that *FOXF1* and *HOXB6* were early positive targets of HAND1, and the activity of their regulons overlapped, whereas the regulon of SOX17 overlapped with the HAND1[−] regulon. These patterns were disrupted in the HAND1-null, while the patterns of endoderm-restricted FOXA2 and HNF4A activity were unchanged. To overcome its absence, a regulon controlled by the cardiac mesoderm master regulator MESP1 was defined using our co-expression data from SCENIC and published RNA-seq data from Mesp1 overexpression (Lin et al, 2022). *HAND1* was confirmed as a MESP1 target in our system. The activity of the MESP1 regulon was not significantly changed by HAND1, suggesting that MESP1 mostly acts upstream and independently of HAND1.

HAND1 binding and increased chromatin accessibility was evident at the *FOXF1* locus and HOXB cluster (Fig. 5H,I). Accessibility at the *FOXF1* promoter was robustly increased later, as seen in the high-HAND1 populations in our day 7.5 ATAC-seq data (Appendix Fig. S2H). A total of 48 genes were common targets of HAND1, FOXF1, and HOXB6, including the genes encoding these upstream regulators as well as other HOXB TFs, suggesting positive feedback and cross-activation. Common targets also included the TFs *LEF1*, *ID2*, and *TEAD1*. Targets specific to FOXF1 and HOXB6 included *JDP2*, *MSX1/2*, *PRDM6*, and *PRRX1*, the activities of which were all increased in the FHF/JCF-like lineage, fibroblasts or epicardial cells (Figs. 4G and EV1). In line with the potential of HAND1 to promote extraembryonic mesoderm specification, we identified HAND1 binding at the *CDX2* locus, but only in the high-HAND1 condition (Fig. 5J). *CDX2* was also a predicted target of FOXF1. The above data allowed us to construct a GRN model of how HAND1 could regulate FHF/JCF and extraembryonic lineage assignment (Fig. 5K).

## HAND1-expressing progenitors are expandable and unique in their multipotency for cardiomyocytes and epicardial cells

Markers indicating the developmental potential of cardiac pro-genitors derived from hPSCs are badly needed to reduce heterogeneity in disease modelling and to develop cell therapies (Zhang et al, 2024). Because the level of HAND1 expression was seemingly indicative of developmental lineage and fate propensity, as illustrated in Fig. 6A, we addressed whether it could be used to distinguish progenitor types and identify and isolate multipotent JCF-like progenitors. To test this, we introduced a *HAND1-T2A-Tomato* knock-in reporter into the *NKX2-5-GFP* hESC line to create a double reporter (Appendix Fig. S6A). HAND1-Tom expression became evident by day 4 of differentiation and was increased by SB and decreased by DMH1 treatment, in agreement with the level of HAND1 by Western blotting (Figs. 5A and 6B). Cardiac progenitors are a transient population in this model and differentiate after little self-renewal. Aiming to circumvent uncontrolled differentiation we examined our GRN data for correlated regulators of progenitor self-renewal and identified MYC- and N-MYC-controlled regulons in mesoderm and early cardiac progenitors (Fig. 6C). The activity of these two regulons declined simultaneously with differentiation towards

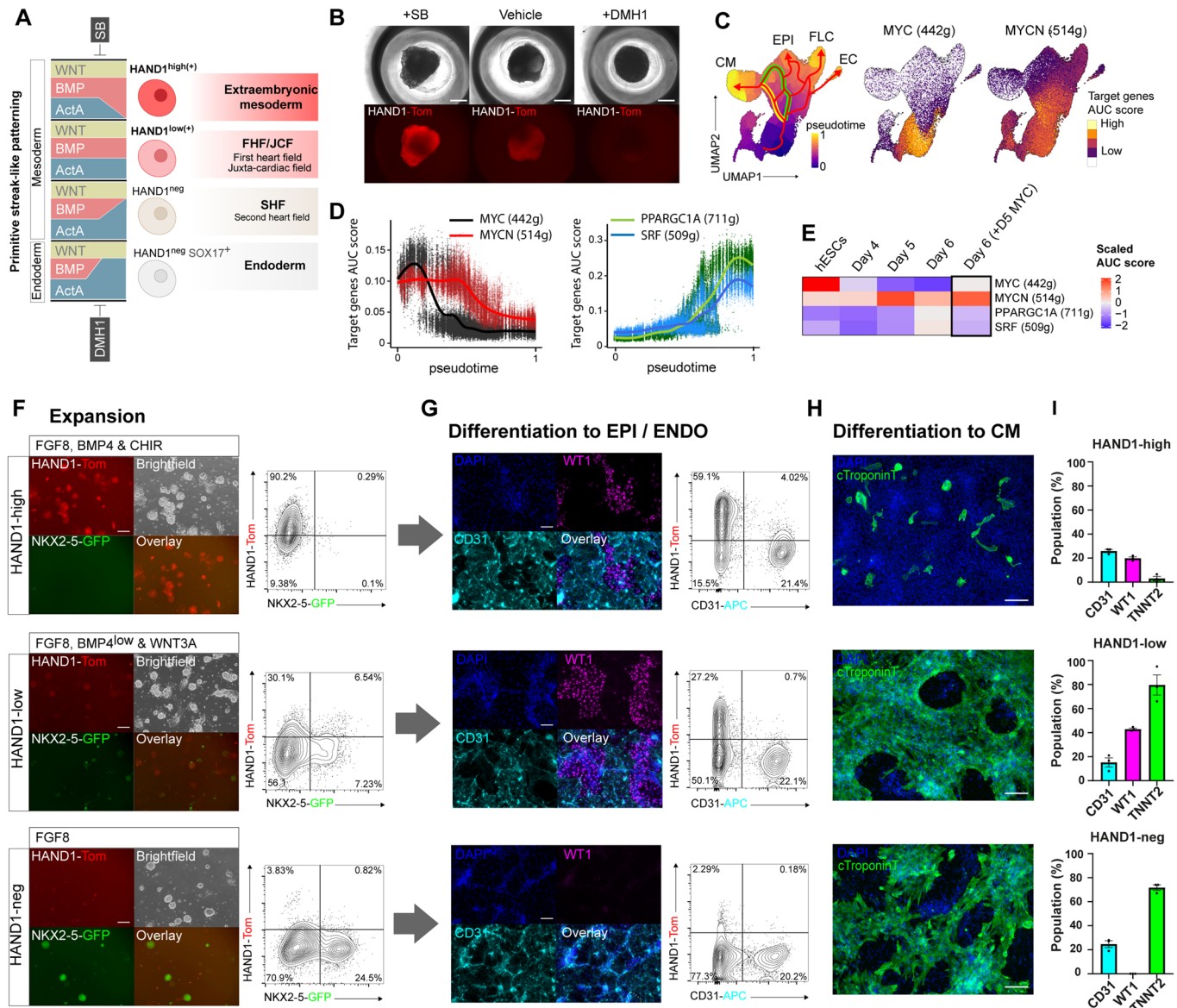

**Figure 6. HAND1-expressing progenitors are expandable and unique in their multipotency for cardiomyocytes and epicardial cells.**

(A) Schematic illustration of HAND1 level with primitive streak-like patterning of hESCs into cardiac and extraembryonic mesoderm. (B) Live images (brightfield and fluorescence) showing HAND1-Tomato in EBs at day 5 of differentiation. (C) UMAP plots showing the target genes AUC score (regulon activity) for MYC and N-MYC with differentiation in wild-type cells. White indicates the score is below threshold. (D) Target genes AUC score for MYC, N-MYC, PGC-1α (PPARGC1A) and SRF through pseudotime of mesoderm differentiation. (E) Scaled target gene AUC scores in bulk populations by differentiation day and impact of induction of a dox-inducible MYC transgene at day 5–6. (F) Expansion of HAND1-high, HAND1-low, and HAND1-neg progenitors from SB, control and DMH1-treated conditions, respectively. HAND1-high progenitors were maintained in FGF8, BMP4, and CHIR; HAND1-low progenitors were maintained in FGF8, BMP4, and WNT3A, and HAND1-negative progenitors were maintained in FGF8-only (see Methods). (G) Differentiation of progenitors to epicardial and endothelial cells assessed by immunostaining for WT1 and CD31, respectively, and additionally by the flow cytometric analysis of CD31. (H) Differentiation of progenitors to cardiomyocytes assessed by immunostaining for cTroponin T. (I) Quantification of differentiated cell types by population. Data in (I) represent mean ± SEM, n = 3 independent biological experiments. Scale bars represent 150 μm. AUC area under the curve, EB embryoid body, Epi epicardial cells, Endo endothelial cells, CM cardiomyocyte, SEM standard error of the mean. See also Appendix Fig. S6. Source data are available online for this figure.

cardiomyocytes, which displayed an increased activity of PGC-1α (*PPARGC1A*) and SRF (Fig. 6D). Both MYC and N-MYC have been shown to play a role in cardiac progenitor expansion in development and in vitro (Birket et al, 2015; Charron et al, 1992; Davis et al, 1993; Muñoz-Martín et al, 2019). A dox-inducible *MYC* transgene activated from day 4.75 was able to halt differentiation

and maintain the activity of the MYC and MYCN regulons (Fig. 6E). This enabled the clonal expansion of these cells supported by IGF-1 and a Hedgehog pathway activator, as previously described (Birket et al, 2015). BMP4, FGF8 and the WNT pathway activators CHIR and WNT3A all promoted HAND1-Tom expression (Appendix Fig. S6B–F). FGF and BMP signals also acted

cooperatively to promote the induction of *NKX2-5* expression, whereas WNT signals did not. To maintain progenitors with HAND1-Tom-high, Tom-low, and Tom-neg statuses, generated in SB-, vehicle- and DMH1-treated EBs, respectively, we optimized three maintenance conditions. The dependence on *MYC* transgene expression remained. Following expansion for 6 days, fluorescence was measured by flow cytometry (Fig. 6F). HAND1-Tom level correlated with a resistance to the induction of NKX2-5-GFP expression, which emerged largely from HAND1-Tom-low or Tom-neg cells. The expanded populations were further differentiated by removing dox and providing specification signals. Differentiation to endothelial and WT1+ mesothelial/epicardial cells was promoted by low-level FGF plus VEGF (Fig. 6G), whereas cardiomyocyte differentiation was promoted by high-level FGF plus TGFβ (Fig. 6H). All populations could generate CD31+ endothelial cells, whereas WT1+ cells were generated exclusively from HAND1-Tom-high and Tom-low cells. HAND1-Tom-high progenitors were unable to make cardiomyocytes, whereas the other populations did so efficiently. HAND1-Tom-low progenitors were multipotent and able to make all three cell types with good efficiency suggesting that they represent a JCF-like population (Fig. 6I).

## Discussion

Through a systematic study of cell diversity in a human model of early cardiac development, we have deciphered lineage trajectories and identified GRNs controlling cell identity and fate. Focusing on the decision point in mesoderm when progenitors may follow either a cardiac FHF/JCF, SHF, or an extraembryonic path, we have uncovered a fate-determining role for the bHLH factor HAND1. The recent seminal discovery of the JCF, a progenitor population marked by *Hand1*, which contributes to both cardiac and extraembryonic tissues including to the epicardial lineage, underpins the developmental significance of these results (Tyser et al, 2021a; Zhang et al, 2021). We found that HAND1 is a master regulator of JCF progenitor specification in mesoderm; in our 3D differentiation model its knockout caused a switch in fate towards a SHF-like lineage. HAND1 restructures chromatin and initiates GRNs including those controlled by HOX and FOXF1. At high levels HAND1 drives an extraembryonic differentiation trajectory. Importantly, HAND1-low progenitors can be expanded and demonstrate a unique multipotency to give rise to both cardiomyocytes and epicardial cells.

We confirmed that the duration of BMP and Activin signalling was a major factor in programming and patterning endoderm and mesoderm in differentiation, as reported in development (Loh et al, 2014; Teague et al, 2024; Hadas et al, 2024). Early manipulation of Activin and BMP signalling in EBs has previously been shown to modulate specification into FHF and SHF-derivatives (Liu et al, 2023; Yang et al, 2022). We further queried why cardiomyocyte development is inhibited or delayed in mesodermal progenitors exposed to a high BMP signalling environment, which programmes progenitors into making more epicardial and fibroblast-like cells. The close proximity of the FHF/JCF and the extraembryonic mesoderm, and overlap in their molecular profiles, makes this question non-trivial (Tyser et al, 2021a). In our system, the expression of *GATA4*, *TBX5*, *MEIS1*, and *PDGFRA* failed to

distinguish closely related progenitors fated to, or not to, become cardiomyocytes. However, by analysing differentially accessible chromatin we deduced that, whereas GATA was universally important, the TFs MEIS, MEF2, NKX2-5, TBX20, and CTCF, presumably acting with some cooperativity, were distinctive in shaping the chromatin structure of cells becoming cardiomyocytes. Conversely, the TFs TEAD, HOX, WT1, and HAND1 were predicted to play the corresponding role in cells not progressing to cardiomyocytes. Focusing on the latter, the function of the Hippo pathway TEAD-YAP1 was found to be important in cardiac progenitor proliferation and survival, consistent with the pathway's known importance in controlling proliferation and organ size in development and survival in PSC differentiation (Croci et al, 2017; LeBlanc et al, 2018). WT1 knockout did not affect cardiomyocyte differentiation, or the inhibitory effect of high BMP signalling levels, thus not supporting a major function for WT1 in human cardiac progenitor formation or differentiation as reported in mouse and zebrafish (Marques et al, 2022; Martínez-Estrada et al, 2010). The enrichment of motifs for HOX TFs was unexpected as their role in cardiac progenitors has been little studied. However, *Hoxb1* is required to maintain proliferation in the posterior SHF in mice (Stefanovic et al, 2020) and was observed in the JCF (Tyser et al, 2021a). The HOXB1–3-null showed that HOX TFs may play a significant role in restricting cardiac differentiation in a high BMP signalling environment. This is also consistent with the effect observed when Hoxb1 is mis-expressed in the mouse anterior SHF or during in vitro differentiation (Stefanovic et al, 2020; Darieva et al, 2025). Finally, HAND1 was found to be an activator of HOX and the major factor in interpreting the response to increased BMP signalling, and promoting epicardial and fibroblast differentiation over cardiomyocytes.

Single cell RNA-seq showed that, in response to modulation of only BMP and Activin signal duration, wild-type cells followed multiple developmental trajectories into FHF/JCF-, extraembryonic mesoderm-, and SHF-like lineages. Interestingly, the fibroblast-like cells, which were generated in large number in the high BMP condition, did not come from the epicardial cells but instead directly from earlier progenitors; some from *HAND1* + *TBX5*-neg *CDX2*-neg cells and others from *HAND1* + *CDX2*+ cells which may represent an extraembryonic mesoderm-like population. Despite some potential compensation by HAND2 (which was expressed in ~10% of mesoderm in the HAND1-null mesoderm), mirroring observations in Hand1-null mice (Firulli et al, 2010; Morikawa and Cserjesi, 2004), the loss of HAND1 resulted in the cells preferentially committing to a SHF-like lineage, suggesting that HAND1 assigns cells to the other lineages—consistent with conclusions from other models (Guo et al, 2024). The validity of these trajectories was supported by our comprehensive GRN analysis. The recent seminal discovery of a shared cardiac- extraembryonic- progenitor marked by *Hand1* has raised the question of what function this bHLH TF has in the formation or differentiation capacity of multipotent progenitors (Tyser et al, 2021a; Zhang et al, 2021). Understanding the developmental functions of Hand1 has been complicated by its important early role in placental development (Firulli et al, 1998; Riley et al, 1998). However, using conditional mutants, a function in cardiac cell proliferation has been described (Firulli et al, 2020; Risebro et al, 2006; Vincentz et al, 2017), and it has been linked to congenital heart disease (Li et al, 2017).

Using a hESC model, free of complications related to extraembryonic dependence, we found that HAND1 could induce a rapid change in chromatin accessibility in mesoderm. Working

with putative binding partners including GATA6, ZIC2/3, FOXO1, FOXF1, PBX/MEIS, and SMAD, our results suggest it plays a major role in activating GRNs defining the FHF/JCF and extraembryonic mesoderm, while inhibiting endoderm differentiation. *HOXB3*, *HOXB6* and *FOXF1* were found to be early HAND1 target genes and inferred to be central components of the GRN directed by HAND1, along with other TFs including MSX1. More work is needed to fully understand the functions of these downstream factors. In our partner manuscript, we show that HOX can inhibit cardiomyocyte differentiation by partnering with MEIS1 and preventing it from binding with GATA to drive differentiation (Darieva et al, 2025). At high levels, HAND1 bound and activated *CDX2* expression. CDX2 is known to promote the development of extraembryonic mesoderm from hPSCs and inhibit cardiac differentiation, as we observed in this condition (Bulger et al, 2024; Rao et al, 2016). A dosage dependency at some regulatory elements but independence at others, is consistent with the behaviour of other TFs functioning in development (Naqvi et al, 2023; Kathiriya et al, 2021; Hannon et al, 2017). *Hand1* heterozygote mutant mice had no apparent phenotype but mice with a *Hand1* hypomorphic allele with 30–40% mRNA expression were embryonic lethal with poor placental function (Firulli et al, 1998; Riley et al, 1998; Firulli et al, 2010). These data suggest that some buffering can take place but there is a sensitivity to Hand1 dosage in development, which should now be closely studied at the protein level and may have implications for developmental defects.

Positioning HAND1 in the developmental hierarchy, our data confirmed that HAND1 lies immediately downstream of the bHLH TF MESP1 (Lin et al, 2022), one of the earliest regulators of mesoderm development (Ajima et al, 2021). They reinforce each other's expression and have many shared target genes including *SNAI2* to promote EMT, and the ability to repress *SOX17* and endoderm development. However, according to the MESP1 ChIP-seq data of Liang et al, they appear to share a limited number of binding sites (14% at low HAND1 and 20% at high HAND1 levels) and their co-binding does not correlate with HAND1-mediated chromatin activation (Liang et al, 2022). This suggests that MESP1 functions more widely to programme mesoderm and HAND1 specifically patterns it to promote FHF/JCF and extra-embryonic fates. The additional promotion of EMT and migration provided by HAND1 is consistent with JCF progenitors being the most highly migratory (Dominguez et al, 2023).

Finally, with regenerative medicine in mind, and building on our previous work (Birket et al, 2015; Schwach et al, 2020), we took advantage of the ability of MYC and N-MYC to promote cell self-renewal in cardiac development to study progenitor fate as a function of HAND1 level. Each progenitor type required a carefully tuned signalling environment to maintain its identity. WNT3A, FGF8, and BMP4 could support the self-renewal of HAND1-low progenitors, which proved to be multipotent for cardiomyocytes and epicardial cells. This provides further evidence of shared lineage at this stage of development. The continued regulation of HAND1 expression by BMP signalling is consistent with the gradient of BMP levels, and *Hand1* mRNA expression, in mesoderm along the mediolateral embryonic axis, extending into extraembryonic mesoderm (Tyser et al, 2021a; Zhang et al, 2021; Hadas et al, 2024; Tonegawa et al, 1997). With the added potential to make endothelial cells, these multipotent progenitors are of

significant interest for cardiac regeneration where all three cell types have been shown important for tissue repair (Bargehr et al, 2019; Cheng et al, 2023; Liu et al, 2018).

In summary, this study provides a comprehensive map of human cardiovascular development and builds on work in the field that has described BMP signalling as a key determinant of mesoderm patterning and heart lineage development (Lee et al, 2017; Teague et al, 2024; Hadas et al, 2024; Liu et al, 2023; Yang et al, 2022). It further reveals important regulatory mechanisms, most notably a lineage-determining function for HAND1 and its role as a major driver of lineage heterogeneity in 3D cardiac differentiation. Controlling these events is crucial for achieving accurate differentiation and maximizing the opportunities of hPSCs for translational applications. These findings will particularly support the modelling of congenital heart disease and the development of cell-based therapies.

# Methods

**Reagents and tools table**

| Reagent/Resource | Reference or Source | Identifier or Catalog Number |
|---|---|---|
| **Experimental models** | | |
| NKX2-5^eGFP/w Human Embryonic Stem Cell Line (HES3) | Elliott Lab | https://doi.org/10.1038/nmeth.1740 |
| NKX2-5-GFP Human Embryonic Stem Cell Line (UMANe002-A-1) (MAN13) | Made in-house Douglas et al, 2024 (https://doi.org/10.1016/j.scr.2023.103262) | https://hpscreg.eu/cell-line/UMANe002-A-1 |
| **Recombinant DNA** | | |
| pSOX17-T2A-dTom-PGK-Neo | This study | Addgene ID 223190 |
| pFOXA2-T2A-mTagBFP2-PGK-Neo | This study | Addgene ID 223191 |
| pHAND1-T2A-dTom-PGK-Neo | This study | Addgene ID 223190 |
| pHAND1-AM-Tag-PGK-Neo | This study | Addgene ID 223216 |
| pENTR4-FLAG | This study | Addgene ID 17423 |
| FLAG-HAND1-T2A-mTagBFP2 | This study | Addgene ID 223192 |
| TO-MYC | Hochedlinger Lab | Addgene ID 19775 |
| pCAG-CRE-IRES2_GFP | Chenn Lab | Addgene ID 26646 |
| FU-tetO-Gateway | Gearhart Lab | Addgene ID 43914 |
| pMDLgp/pRRE | Trono Lab | Addgene ID 12251 |
| pMD.G | Trono Lab | Addgene ID 12259 |
| pRSV-Rev | Trono Lab | Addgene ID 12253 |
| FUdeltaGW-rtTA | Hochedlinger Lab | Addgene ID 19780 |

| Reagent/Resource | Reference or Source | Identifier or Catalog Number |
|---|---|---|
| **Antibodies** | | |
| Polyclonal Goat anti-HAND1 (1 µg/mL) | R & D Systems | #AF3168 |
| Monoclonal Mouse anti-PDGFRα-BV421 (1:50) | BD Biosciences | #526799 |
| Monoclonal Mouse anti-CD235a-APC (1:50) | BD Biosciences | #561775 |
| Monoclonal Mouse anti-ACTN2 (1:800) | Sigma-Aldrich | #A7811 |
| Monoclonal Rabbit anti-WT1 (1:200) | Abcam | #ab89901 |
| Monoclonal Mouse anti-CD31 (1:50) | ThermoFisher Scientific™ | #BMS137 |
| Monoclonal Mouse anti-FOXA2 (1:100) | BD Biosciences | #561580 |
| Monoclonal Mouse anti-CD13-APC (1:100) | BioLegend | #301705 |
| Polyclonal Rabbit anti-Cardiac Troponin T (1:800) | Proteintech | #15513-1-AP |
| Monoclonal Rabbit anti-YAP (1:500) | Abcam | #ab205270 |
| Polyclonal Goat anti-SOX17 (1:200) | R & D | #AF1924 |
| Mouse AbFlex® AM-Tag (rAb) (10 µg/ChIP) | Active Motif | #91111 |
| Polyclonal Alexa Fluor™-488 Donkey Anti-Rabbit IgG (1:500) | Invitrogen | #A21206 |
| Polyclonal Alexa Fluor™-555 Donkey Anti-Mouse IgG (1:500) | Invitrogen | #A31570 |
| Polyclonal Alexa Fluor™-594 Donkey Anti-Mouse IgG (1:500) | Invitrogen | #A21203 |
| Polyclonal Alexa Fluor™-594 Goat Anti-Rabbit IgG (1:500) | Invitrogen | #A-11012 |
| Polyclonal Alexa Fluor™-488 Donkey Anti-Goat IgG (1:500) | Invitrogen | #A-11055 |
| Monoclonal Rabbit anti-SMAD1 (1:1000) | Cell Signalling Technology | #6944 |
| Monoclonal Rabbit anti-Phospho-SMAD1/5/9 (1:1000) | Cell Signalling Technology | #13820 |
| Monoclonal Rabbit anti-SMAD2 (1:1000) | Cell Signalling Technology | #5339 |
| Monoclonal Rabbit anti-Phospho-SMAD2 (1:1000) | Cell Signalling Technology | #3108 |
| HRP-linked anti-Rabbit IgG (1:10,000) | Cell Signalling Technology | #7074 |
| Polyclonal HRP-linked Rabbit Anti-Goat IgG (1:10,000) | Abcam | #ab97100 |
| Monoclonal HRP-linked Rabbit Anti-GAPDH (1:10,000) | Abcam | #ab204481 |

| Reagent/Resource | Reference or Source | Identifier or Catalog Number |
|---|---|---|
| **Oligonucleotides and other sequence-based reagents** | | |
| gRNAS for CRISPR knock-in | This study | Appendix Table S1 |
| gRNAS for CRISPR knockout | This study | Appendix Table S2 |
| RT-qPCR Primers | This study | Appendix Table S4 |
| Barcoding primers for ATAC-seq | ATAC-Seq Kit (Active Motif) | Appendix Table S5 |
| **Chemicals, Enzymes and other reagents** | | |
| Activin A | Miltenyi Biotech | 130-115-009 |
| BMP4 | R & D Systems | 314-BP-050 |
| CHIR99021 | Selleckchem | S1263 |
| XAV939 | VWR | CAYM13596-1 |
| Doxycycline | Sigma | D9891-1G |
| SAG | Selleckchem | S7779-SEL-2mg |
| LONG R3 IGF-1 | Sigma | 85580C |
| SB431542 | Generon | A10826-10 |
| DMH1 | Merck | D8946-5MG |
| bFGF | Miltenyi Biotech | 130-104-924 |
| TGFβ1 | Peprotech | 100-21 |
| WNT3A | Proteintech | HZ-1296 |
| Dexamethasone | Merck | S2915-100MG |
| T3 | Merck | T6397-100MG |
| VEGF165 | Proteintech | HZ-1038 |
| IGF-1 | Sigma | 85580C |
| Thiazovivin | Cambridge Biosciences | SM35-2 |
| Alt-R™ S.p Cas9 Nuclease V3 | IDT | 1081058 |
| DAPI | Merck | D9542-10MG |
| 7-AAD | Invitrogen™ | A1310 |
| DRAQ7™ Dye | Invitrogen™ | D15106 |
| Corning® Matrigel® Growth Factor Reduced (GFR) Basement Membrane Matrix, LDEV-free | Corning® | 354230 |
| DMEM/F12 | Gibco | 21331020 |
| KnockOut™ Serum Replacement | Gibco | 10828-028 |
| Non-essential amino acids | Gibco | 11140035 |
| GlutaMAX™ | Gibco | 35050038 |
| β-mercaptoethanol | Gibco | 31350010 |
| Penicillin-Streptomycin | Sigma | P0781-20ML |
| Foetal bovine serum (FBS) | Gibco | 10500064 |
| DMEM- High Glucose | Sigma | D6429-500ML |
| IMDM | Gibco | 21056023 |
| F12 | Gibco | 31765027 |
| PFHM II | Gibco | 12040-077 |

| Reagent/Resource | Reference or Source | Identifier or Catalog Number |
|---|---|---|
| Bovostar | CatusBiotech GmbH | BSAS 0.1 |
| PVA | Sigma | P8136 |
| CD Lipids | Gibco | 11905031 |
| Insulin-Transferrin-Selenium-Ethanolaminie (ITS-X) | Gibco | 51500056 |
| 1-Thioglycerol | Sigma | M6145 |
| L-Ascorbic Acid 2-phosphate | Sigma | A8960-5G |
| Phenol Red Solution | Sigma | P0290-100ML |
| TryPLE™ Select Enzyme (1X) | Gibco | 12563011 |
| TryPLE™ Select Enzyme (10X) | Gibco | A1217701 |
| Alt-R® CRISPR-Cas9 tracrRNA | IDT | 1072532 |
| Alt-R® Cas9 Electroporation Enhancer | IDT | 1075915 |
| Neomycin | Sigma | N1142 |
| Lipofectamine™ 2000 | Invitrogen™ | 11668019 |
| Lenti-X 293T Cells | Takara Bio | 632180 |
| MEF Feeder Cells | Cell Biolabs | CBA-310 |
| Pierce™ BCA protein assay | ThermoFisher Scientific™ | 23225 |
| ECL™ Prime Western Blotting Detection Reagent | Amersham™ | RPN2232 |
| MiniPROTEAN TGX Stain-free gels 4–12% | Bio-rad | 4568095 |
| TransBlot® Turbo™ Mini-Size Nitrocellulose | Bio-rad | 1704158 |
| Trans-Blot Turbo 5x Transfer Buffer | Bio-rad | 10026938 |
| Halt™ Protease and Phosphatase Inhibitor Cocktail | ThermoFisher Scientific™ | 78440 |
| Halt™ Protease Inhibitor Cocktail | ThermoFisher Scientific™ | 87786 |
| RIPA Buffer | ThermoFisher Scientific™ | 89900 |
| Tetro™ cDNA Synthesis Kit | Meridian Bioscience® | BIO-65042 |
| Power SYBR™ Green Master Mix | Applied Biosystems™ | 4367659 |
| PureLink™ RNA Mini Kit | Invitrogen™ | 12183018A |
| TRIzol™ Reagent | Invitrogen™ | 15596026 |
| Dynabeads™ Protein G | Invitrogen™ | 10004D |
| Human Stem Cell Nucleofector® Kit 2 | Lonza | VPH-5022 |
| ATAC-Seq Kit | Active Motif | 53150 |
| MyFi™ DNA Polymerase | Bioline | BIO-21117 |
| Q5® High-Fidelity DNA Polymerase | New England Biolabs® | M0491S |
| NEBuilder® HiFi DNA Assembly Master Mix | New England Biolabs® | E2621S |
| QIAamp® DNA Mini Kit | QIAGEN | 51304 |
| Gateway™ LR Clonase™ II Enzyme mix | Invitrogen™ | 11791020 |

| Reagent/Resource | Reference or Source | Identifier or Catalog Number |
|---|---|---|
| Single Cell Fixed RNA Sample Preparation Kit | 10X Genomics | 1000414 |
| ProLong™ Gold Antifade Mountant | Invitrogen™ | P10144 |
| One Shot™ Stbl3™ Chemically Competent Cells | Invitrogen™ | C737303 |
| NEB® 5-alpha Competent *E. coli* (High Efficiency) | New England Biolabs® | C2987H |
| 6-well plates | Corning® | BC010 |
| 96-well plates (flat bottom) | Corning® | BC015 |
| 96-well plates (V bottom) | Grenier | 651161 |
| 96-well plates lids | Grenier | 656161 |
| **Software** | | |
| FlowJo (v10) | | https://www.flowjo.com/ |
| GraphPad Prism 9 | | https://www.graphpad.com/ |
| R Studio (v4.0.5) | | https://posit.co/download/rstudio-desktop/ |
| Galaxy | | https://usegalaxy.org/ |
| Adobe Illustrator 2024 | | https://www.adobe.com/uk/products/illustrator.html |
| **Other** | | |
| QuantStudio 1 Real-Time PCR System | ThermoFisher Scientific™ | |
| 7900HT Fast Real-Time PCR System | Applied Biosystems™ | |
| Bioruptor® | Diagenode | |
| Illumina NovaSeq6000 Platform | Illumina | |
| ChemiDoc™ Imaging System | Bio-rad | |
| Trans-Blot Turbo Transfer | Bio-rad | |
| Leica SP8 Confocal Microscope | Leica | |
| BD LSRFortessa™ Cell Analyzer | BD Biosciences | |
| BD Influx™ Cell Sorter | BD Biosciences | |
| BD FACSAria™ Fusion Flow Cytometer | BD Biosciences | |
| Evos™ M5000™ Imaging System | ThermoFisher Scientific™ | |

## Methods and protocols

### Maintenance of human ESC lines

HES3 *NKX2-5*^*eGFP/w*^ hESCs (Elliott et al, 2011) and gene-edited derivatives were maintained on a layer of mitotically-inactivated mouse embryonic fibroblasts (MEFs) in DMEM/F12-based medium containing 20% (v/v) KnockOut™ Serum Replacement (Gibco), 100 mM non-essential amino acids (Gibco), 2 mM GlutaMAX™

(Gibco), 0.1 mM β-mercaptoethanol (Gibco) and 10 ng/ml bFGF (Miltenyi Biotech). hPSCs were passaged using TrypLE Select (ThermoFisher Scientific™) every 3–4 days. In replicate experiments, UMANe002-A-1 *NKX2-5-GFP* hESCs (Douglas et al, 2024) and gene-edited derivatives were used and were maintained in the same way. Both lines were authenticated by STR analysis and confirmed to be free of mycoplasma contamination.

### Gene editing by CRISPR-Cas9

Fluorescent reporters for *SOX17* (dTomato), *FOXA2* (mTagBFP2) and *HAND1* (dTomato) were generated using a 3' gene targeting strategy to incorporate the fluorophore downstream of a T2A peptide. C-terminal epitope-tagged *HAND1* was generated using a targeting strategy to incorporate an AM-Tag (Active Motif). Targeting plasmids incorporated 700–800 bp homology arms and a loxP-flanked neomycin resistance cassette and were assembled using NEBuilder HiFi Assembly (NEB). The plasmids have been deposited as follows: pSOX17-T2A-dTom-PGK-Neo (Addgene ID 223190), pFOXA2-T2A-mTagBFP2-PGK-Neo (Addgene ID 223191), pHAND1-T2A-dTom-PGK-Neo (Addgene ID 223190), pHAND1-AM-Tag-PGK-Neo (Addgene ID 223216). We recently described the generation of a *HAND1-T2A-Tomato* reporter using the same method (Lynch et al, 2024). In brief, ribonucleoprotein (RNP) complexes of CRISPR-Cas9 were made by combining 120 pmol of crRNA:tracrRNA mix (IDT) with 67 pmol of Cas9 (Alt-R™ S.p. Cas9 Nuclease V3, IDT), final concentrations 1.3 and 1.1 μM, respectively. gRNA sequences are listed in Appendix Table S1. $0.5 \times 10^6$ cells were transfected with the RNP mix, 120 pmol Alt-R® Cas9 Electroporation Enhancer (1.3 μM final) and 1 μg of donor plasmid (pSOX17-T2A-dTom-PGK-Neo, pFOXA2-T2A-mTagBFP2-PGK-Neo pHAND1-T2A-dTom-PGK-Neo or pHAND1-T2A-Am-Tag). Transfection was performed by electroporation using an Amaxa Nucleofector II combined with Human Stem Cell Nucleofector Solution 2 (Lonza, ref #: VPH-5022) and programme B-16. Cells were selected with 100 μg/ml neomycin for 5 days and then electroporated with 1 μg of pCAG_-CRE_IRES_GFP plasmid (Addgene 26646) expressing Cre recombinase and GFP to excise the neomycin selection cassette. 48 h later, single GFP+ cells were sorted by FACS (BD Influx) into MEF-coated 96-well plates. DNA was isolated from clones using a Qiagen DNA Mini kit for PCR screening using primers spanning the limits of the homology arms. PCRs were performed using Q5 High-Fidelity DNA polymerase (NEB) or MyFi™ DNA Polymerase (Bioline), with the supplier's standard protocol. Clones with correct left and right arm integration were further examined for the presence of a targeted or untargeted allele to verify zygosity. Accurate targeting was confirmed by Sanger sequencing.

Gene knockouts (KO) of *YAP1*, *WT1*, *HOXB1-2-3* and *HAND1* were generated by targeting the earliest coding exon of each gene with a pair of gRNAs predicted to generate a frameshift deletion. The *HOXB1-2-3* triple KO line was made by sequential mutation of *HOXB1*, *HOXB2* and *HOXB3*. RNP complexes were generated and delivered as above. gRNA sequences are listed in Appendix Table S2. Three days after targeting, cells were cloned by FACS into MEF-coated 96-well plates. DNA was isolated from clones using a Qiagen DNA Mini kit for PCR screening using primers spanning the predicted deletion. Homozygous mutation was confirmed by Sanger sequencing.

### TO-HAND1 lentiviral vector cloning

The *HAND1* coding sequence (CDS) was obtained commercially (Horizon Discovery; Clone 3162118) and cloned into pENTR4-FLAG

(Addgene ID 17423). A T2A-mTagBFP2 fragment was inserted downstream of the CDS to generate an entry vector containing FLAG-HAND1-T2A-mTagBFP2 (deposited as Addgene ID 223192). The entry vector was recombined with the destination vector FU-tetO-Gateway (Addgene #43914) using Gateway™ LR Clonase™ II Enzyme Mix (ThermoFisher Scientific™) and transformed into One Shot™ Stbl3™ Chemically Competent cells (ThermoFisher Scientific™).

### Lentivirus production and hPSC transduction

The lentiviral packaging vectors (pMDLg.pRRE [Addgene ID 12251]; pMD.G [Addgene ID 12259] and pRSV-Rev [Addgene ID 12253]), together with the transgene-containing destination vector (TO-HAND1 as described above or TO-MYC [Addgene ID 19775]), were transfected into Lenti-X 293T cells (Takara Bio) using Lipofectamine 2000 (ThermoFisher Scientific™). The supernatants were collected daily for 2 days, and the lentiviruses concentrated by ultracentrifugation and stored at −80 °C.

The *TO-HAND1-T2A-BFP* system was introduced by transducing HAND1-null hPSCs with two lentiviruses, one carrying the *HAND1-T2A-mTagBFP2* transgene under the control of a TRE-CMV promoter (described above), and the second carrying the tetracycline transactivator (Addgene ID 19780). A *TO-MYC* system was introduced by transducing the hPSCs with two lentiviruses, one carrying a *MYC T58A* transgene under the control of a TRE-CMV promoter (Addgene ID 19775), and the second carrying the tetracycline transactivator (Addgene ID 19780). Polyclonal populations were used throughout.

### hPSC differentiation

Differentiations were performed in serum-free BPEL medium (Ng et al, 2008) containing 1 μg/ml insulin using an embryoid body (EB) system. EBs were formed by depositing 2500 cells in 50 μl differentiation medium per well in V-bottomed 96-well plates (Greiner). The following growth factors were present for the first 3 days of differentiation: 25 ng/ml BMP4 (R&D Systems), 25 ng/ml Activin A (Miltenyi Biotech) and 1.5–1.75 μM CHIR99021 (Selleckchem). Where indicated, the ALK5 inhibitor SB431542 (Generon), ALK2/3 inhibitor DMH1 (Merck), or vehicle-only (DMSO), were added on day 2 at the specified concentration as a 2 μl bolus and mixed. On day 3, wells were refreshed with 100 μl BPEL. On day 6, wells were refreshed with 100 μl BPEL supplemented with 100 pg/ml bFGF (Miltenyi Biotech). On day 9, wells were refreshed with 100 μL BPEL. Where indicated, on day 7.5, cells were dissociated with 1X TrypLE Select and sorted by Tomato and GFP fluorescence using FACS (BD Aria) for RNA- and ATAC-seq. DAPI was used as a viability dye. In replicate experiments using UMANe002-A-1 *NKX2-5-GFP* hESCs, 1 μM XAV939 (VWR) was included between day 3 and 6 of EB differentiation which improves cardiomyocyte differentiation efficiency in this line.

Doxycycline-inducible HAND1 (*TO-HAND1-T2A-BFP*) hESCs were differentiated in monolayers in BPEL plus growth factors, as described above. On day 2.5, 1 μg/ml doxycycline was added for 12 h. On day 3, cells were dissociated with 1X TrypLE Select and sorted by BFP level (negative, medium and high) using FACS (BD Influx) for RNA- and ATAC-seq. 3 μM DRAQ7™ was used as a viability dye.

### Progenitor cell maintenance and differentiation

For progenitor culture of *TO-MYC* cells. On day 4.75, doxycycline was added to EBs at 1 μg/ml for 24 h. After dissociation with 1X TrypLE

Select, progenitors were seeded on a thin layer of undiluted Matrigel (14 μl/cm²) (growth factor reduced; Corning 354230) in organ culture dishes (BD Falcon) at a cell density of $8 \times 10^3$/cm² in BPEL medium containing 1 μg/ml doxycycline, 5 μM SB431542, 100 ng/ml LONG R3 IGF-1 (hereafter, IGF-1) and 1 μM SAG (Selleckchem) plus additional supplements as indicated: 5–100 ng/ml FGF8b (Proteintech), 0.5–25 ng/ml BMP4 (R&D Systems), 200 ng/ml WNT3A (Proteintech) and 1 μM CHIR99021 (Selleckchem). Cultures were refreshed every 2 days.

Cardiomyocyte differentiation was performed by seeding cells at a density of $2 \times 10^5$ cells/cm² on Matrigel-coated (1:100 dilution) 96-well plates (BD Falcon) in BPEL medium supplemented with 100 ng/ml IGF-1, 1 μM SAG, 1 ng/ml bFGF and 2.5 ng/ml TGFβ1 (Peprotech). On day 2, medium was changed to BPEL supplemented with 100 ng/ml IGF-1, 100 pg/ml bFGF and 2.5 ng/ml TGFβ1. On day 4, medium was changed to BPEL supplemented with 100 ng/ml IGF-1 and 10 pg/ml bFGF. On day 6, medium was changed to BPEL supplemented with 100 ng/ml IGF-1, 100 nM T3 (Merck) and 1 μM Dexamethasone (Merck). Cells were fixed for immunocytochemistry on day 8.

Endothelial and epicardial differentiation was performed by seeding cells at a density of $6 \times 10^4$ cells/cm² on Matrigel-coated (1:100 dilution) 96-well plates (BD Falcon) in BPEL medium supplemented with 5 μM SB431542, 100 ng/ml IGF-1, 50 ng/ml VEGF165 (Proteintech) and 50 pg/ml bFGF. This medium was refreshed at day 2, 4 and 6. On day 8, cells were dissociated with 2X TrypLE Select for flow cytometric analysis or fixed for immunocytochemistry.

### Flow cytometry and sorting

EBs were dissociated into single cells with TrypLE Select (1X or 10X depending on the stage). Staining was performed in FACS buffer (0.5% BSA/1 mM EDTA in PBS) with the following antibodies (Appendix Table S3): anti-PDGFRα-BV421 (BD Biosciences 562799), anti-TRA181-FITC (BD Biosciences cat no. 560883), and CD235a-APC (BD Biosciences 561775). Staining was performed at room temperature for 15 min followed by washing with FACS buffer. Measurements were performed using LSRFortessa™ Cell Analyzer (BD Biosciences). A viability dye was added to exclude dead cells (DAPI/DRAQ7/7-AAD, depending on the panel). Cell sorting was performed on a BD FACSAria™ Fusion Flow Cytometer or a BD Influx™ cell sorter (BD Biosciences), with samples maintained at 4 °C throughout. Flow cytometric data was analysed with FlowJo (v10).

### RNA isolation and quantitative reverse transcription PCR (RT-qPCR)

RNA was extracted using TRIzol followed by purification using the Purelink RNA Mini Kit (ThermoFisher Scientific™), according to the manufacturer's instructions. Genomic DNA was removed by on-column DNAse digestion. cDNA was generated from 1 μg of total RNA using the Tetro cDNA Synthesis Kit (Bioline) according to the manufacturer's instructions. qPCR was performed using Power SYBR™ Green PCR Master Mix (ThermoFisher Scientific™). Relative gene expression was obtained by normalising to the housekeeping genes *RPLPO* and *GUSB*. Thermocycling was performed on a 7900HT qPCR system (Applied Biosystems) or QuantStudio 1 (ThermoFisher Scientific™). The ΔΔCt method was used to calculate relative gene expression. Primers are listed in Appendix Table S4.

### Bulk RNA-seq

Quality and integrity of the RNA samples were assessed using a 4200 TapeStation (Agilent Technologies) and then libraries generated using the Illumina® Stranded mRNA Prep Ligation kit (Illumina, Inc.) according to the manufacturer's protocol. Briefly, total RNA (typically 0.025–1 μg) was used as input material from which polyadenylated mRNA was purified using poly-T, oligo-attached, magnetic beads. Next, the mRNA was fragmented under elevated temperature and then reverse transcribed into first strand cDNA using random hexamer primers and in the presence of Actinomycin D (thus improving strand specificity whilst mitigating spurious DNA-dependent synthesis). Following removal of the template RNA, second strand cDNA was then synthesized to yield blunt-ended, double-stranded cDNA fragments. Strand specificity was maintained by the incorporation of deoxyuridine triphosphate (dUTP) in place of dTTP to quench the second strand during subsequent amplification. Following a single adenine (A) base addition, adaptors with a corresponding, complementary thymine (T) overhang were ligated to the cDNA fragments. Pre-index anchors were then ligated to the ends of the double-stranded cDNA fragments to prepare them for dual indexing. A subsequent PCR amplification step was then used to add the index adaptor sequences to create the final cDNA library. The adaptor indices enabled the multiplexing of the libraries, which were pooled prior to cluster generation using a cBot instrument. The loaded flow-cell was then paired-end sequenced (76 + 76 cycles, plus indices) on an Illumina HiSeq4000 instrument. Finally, the output data was demultiplexed and BCL-to-Fastq conversion performed using Illumina's bcl2fastq software (v2.20.0.422). Input sequence was mapped using STAR (v2.7.7a). Gene annotation used was Human Genocode 37 for hg38/GRCh38 Ensembl (v103). Read counts were generated by STAR. Ensembl gene identifiers and gene names were assigned to each gene.

For day 3 analysis, DESeq2 (v3.19) was used to identify genes differentially expressed in either HAND1-BFP^high or HAND1-BFP^med fractions compared to the HAND1-BFP^neg fraction (FDR < 0.05). Genes responding to doxycycline treatment independently of HAND1 were identified and subtracted by performing control experiments in a line not carrying the TO-HAND1 lentivirus. All visualization of RNA-seq analysis was carried out in R Studio (v4.0.5).

### ATAC-Seq

After cell sorting, samples were prepared for ATAC-seq analysis using an ATAC-Seq Kit (Active Motif) as per the manufacturer's instructions. Briefly, cells were gently washed with ice cold PBS and lysed with ATAC lysis buffer. Tagmentation master mix was prepared (Tagmentation buffer, PBS, 0.5% Digitonin, 10% Tween 20, Assembled Transposomes) and tagmentation carried out at 37 °C for 30 min at 800 rpm. DNA was purified and then PCR amplified using unique combinations of i7 and i5 indexing primers per sample. After PCR amplification, samples were purified using SPRI beads and quality determined using Agilent Tapestation 2200. Barcoding primers are listed in Appendix Table S5.

The adaptor indices enabled the multiplexing of the libraries, which were pooled prior to cluster generation using a cBot instrument. The loaded flow-cell was then paired-end sequenced (76 + 76 cycles, plus indices) on an Illumina HiSeq4000 instrument. Finally, the output

data was demultiplexed and BCL-to-Fastq conversion performed using Illumina's bcl2fastq software (v.2.20.0.422).

Briefly, peaks were called using MACS2 (day 7.5 samples) or MACS3 (day 3 samples) as 200 bp regions centred on the MACS peak summit. Overlapping regions were combined to create longer regions. Data were normalized based on sequencing depth. Peak annotation was performed using UpsetR (v1.4.0) and ChIPSeeker (v3.19). For the day 3 samples (HAND1-BFP+ and HAND1-BFPneg), differential accessibility analysis was carried out using DiffBind (v3) and DESeq2 (v3.19) with a default threshold of false discovery rate (FDR) < 0.05.

### Single cell RNA-seq: sample and library preparation

EBs were dissociated to single cells as described above. Cells from different conditions were combined as follows: 25% SB-treated cells, 25% DMH1-treated cells and 50% control (vehicle-only) cells. Samples were collected at day 3, 4, 5, 6, 7, 8 and 10. Additional wild-type replicates at days 7 and 8 were included. A total of $1-2 \times 10^6$ cells were used for each sample. The Single Cell Fixed RNA Sample Preparation Kit (10X Genomics) was used to fix and cryopreserve the cells according to the manufacturer's instructions. In brief, cells were fixed with Fixation Buffer containing 4% formaldehyde (ThermoFisher Scientific™) overnight at 4 °C. 24 h later, cells were quenched with freshly prepared quench buffer and stored at $-80 \, ^\circ$C. Frozen cells were thawed and washed in 0.5X PBS/0.02% BSA (Miltenyi)/0.2 U/ml RNase Inhibitor (Roche). Barcoding was performed using the Chromium Next GEM Single Cell Fixed RNA Human Transcriptome Probe Kit (10X Genomics) according to the manufacturer's instructions. Up to $2 \times 10^6$ cells per timepoint and genotype (wild-type and HAND1-null) were re-suspended in Hybridisation Mix with a single Human WTA barcode (10x Genomics, Inc. Pleasanton, USA) according to the manufacturer's protocol (CG000527 Rev C). Briefly, samples were incubated overnight with probe pairs designed to hybridize to mRNA, samples were pooled and washed to remove unbound probes. Nanoliter-scale Gel Beads-in-emulsion (GEMs) were generated by loading barcoded Gel Beads, a master mix containing pooled, probe-hybridized cells, and partitioning oil onto a Chromium Next GEM chip. Cells were delivered at a limiting dilution, such that the majority (90–99%) of generated GEMs contain no cell, while the remainder largely contain a single cell. The Gel Beads were then dissolved, primers, including an Illumina TruSeq Read 1 sequence, a 16-nucleotide 10x Barcode, a 12-nucleotide unique molecular identifier (UMI) and partial Capture Sequence 1, were released, and any co-partitioned cells lysed.

Following GEM generation, a ligation step sealed the nick between the left- and right-hand probe and subsequently the Gel Bead primer was hybridised to the capture sequence on the ligated probe pair and extended by a polymerase adding the UMI, 10x Barcode and partial Read 1 sequence before a heat-denaturation step inactivated the enzymes. Post-incubation, GEMs were broken and the recovered ligated product pre-amplified before being cleaned up by SPRIselect. Illumina-compatible sequencing libraries were constructed by adding P5, P7, i5 and i7 sample indexes, and Illumina Small Read 2 sequences to the 10x barcoded, ligated probe products via Sample Index PCR followed by a SPRIselect size-selection.

### scRNA-seq analysis: library sequencing

The resulting sequencing library comprised standard Illumina paired-end constructs flanked with P5 and P7 sequences. The 16 bp 10x Barcode and 12 bp UMI were encoded in Read 1, while Read 2 sequenced the ligated probe insert, constant sequence and the 8 bp Probe Barcode. Sample indexes were incorporated as the i5 and i7 index reads. Paired-end sequencing (28:90) was performed on the Illumina NovaSeq6000 platform. The .bcl sequence data were processed for QC purposes using bcl2fastq software (v. 2.20.0.422) and the resulting .fastq files assessed using FastQC (v. 0.11.3), FastqScreen (v. 0.14.0) and FastqStrand (v. 1.13.0) prior to pre-processing with the CellRanger pipeline.

### scRNA-seq analysis: read mapping to genome and cell filtering

Raw sequencing data were processed using the 10x Genomics Cell Ranger pipeline (v7.1.0). Base call (BCL) files generated by the sequencer were converted to FASTQ files using "cellranger mkfastq". The FASTQ files were then mapped against the pre-built human reference package from 10X Genomics (GRCh38-2020-A) using "cellranger multi" to demultiplex and produce the gene-cell barcode matrix for individual samples.

The single-cell data were processed in R environment (v4.1) following the workflow documented in Orchestrating Single-Cell Analysis with Bioconductor (Amezquita et al, 2020). Briefly, for each sample, the HDF5 file generated by Cell Ranger was imported into R to create a SingleCellExperiment object. A combination of median absolute deviation (MAD), as implemented by the "isOutlier" function in the scuttle R package (v1.4.0) and exact thresholds was used to identify and subsequently remove low-quality cells before data integration.

### scRNA-seq: data integration, visualization, cell clustering and annotation

The log-normalized expression values of the combined data were computed using the "multiBatchNorm" function from the batchelor R package (v1.10.0). The per-gene variance of the log-expression profile was modelled using the "modelGeneVarBy-Poisson" function from the scran R package (v1.22.1) and top 5000 highly variable genes were selected. The mutual nearest neighbours (MNN) approach implemented by the "fastMNN" function from the batchelor R package was used to perform batch correction.

For the combined analysis, the first 50 dimensions of the MNN low-dimensional corrected coordinates for all cells were used as input to produce the uniform manifold approximation and projection (UMAP) using the "runUMAP" function from the scater R package (v1.22.0). Putative cell clusters were identified using the Leiden algorithm from the igraph R package (v1.5.1). Cluster-specific markers were identified using the "findMarkers" function from the scran R package, which performs pairwise t-tests between clusters.

### scRNA-seq: lineage trajectory analysis and marker gene expression

Separate wild-type and HAND1-null analyses were performed using Monocle3 (v.1.3.4). The raw counts were pre-processed using the "preprocess_cds" function. The HVGs were as identified from the combined analysis but filtered to include only genes implicated in heart development (Zhou et al, 2023), excluding cell cycle-related genes (from the Seurat package dataset 'cc.genes.updated.2019'). Dimensionality reduction by UMAP and in the examples was performed using a minimum distance of 1 and sqrt(n cells) nearest neighbours. Functions "learn_graph" and "order_cells" were performed using default settings. To find consensus trajectories the UMAP settings were varied, and the

dominant paths were taken. For gene expression plots a minimum expression threshold of 0.7 was used.

### HAND1 target gene identification

For the HAND1-high against HAND1-low cell comparison in the wild-type mesoderm, the HAND1-high population was defined as cells with HAND1 expression in the top quantile, and the HAND1-low population as cells with expression in the bottom quantile. For the wild-type against HAND1-null mesoderm comparison, the mesoderm clusters of both were selected. Differential expression analysis of HAND1-null against wild-type and HAND1-high against HAND1-low in the mesoderm was performed using the "nbTestSH" function (FDR < 0.05) from the sSeq package (Yu et al, 2013). Genes affected concordantly were considered putative HAND1 targets. To generate the final regulons, these targets were further filtered using the HAND1 ChIP-seq data and only genes with a HAND1 binding site within 100 kb of the gene's transcription start site (TSS) were retained. TSS were obtained from the TxDb.Hsapiens.UCSC.hg38.knownGene package (v3.2).

### Gene ontology analysis

Gene ontology (GO) enrichment analysis of differentially expressed genes was performed using the enrichR R package (v3.2). Results were ranked by the combined scores (calculated as log(P-value) biormultiplied by z-score) and terms with an adjusted P-value < 0.05 were considered significant. The GOplot package was used to generate a circularly composited overview of HAND1 target genes and their assigned Biological Processes (2023) GO terms.

### Gene regulatory network analysis by SCENIC

After QC filtering and pre-processing as above, the PySCENIC pipeline was followed as described (Van de Sande et al, 2020). Network inference was performed using the GRNBoost2 algorithm. For motif enrichment analysis and TF-regulon prediction by cisTarget, two ranking databases were used: 'hg38__refseq-r80__10kb_up_and_down_tss.genes_vs_motifs.rankings.feather' and 'encode_20190621__ChIP_seq_transcription_factor.hg38__refseq-r80__10kb_up_and_down_tss.max.genes_vs_tracks.rankings.feather'. The wild-type, cardiac progenitors-only and HAND1-null cells were analysed separately. GRNBoost2 and cisTarget were run 5 times using different seed values and only regulators identified in all 5 runs were retained. Regulon target gene lists were aggregated and filtered to include all genes identified in at least 2 runs. Where unique regulators were identified in the cardiac progenitors or the HAND1-null cells they were added to the final list of regulons. Custom regulons were defined for HAND1 and MESP1 as described in the results. The MESP1 regulon was defined using published RNA-seq data from a Mesp1 overexpression system (Lin et al, 2022). Genes upregulated at 12 h (doxycycline vs control; 'batch 1') were identified using DESeq2 (v3.19) (FDR < 0.05; fold-change > 1.5) and intersected with SCENIC adjacencies in our data to enrich for direct targets. AUCell was performed in R to score the activity of these regulons across all cells. The automatically selected regulon activity thresholds were used for plotting. Regulon specificity scores (RSS) were calculated and those above the 95th percentile marked.

### Constructing a gene regulatory network model for HAND1

A hierarchical gene regulatory network aiming to explain early HAND1-dependent regulatory events was constructed using the data from SCENIC together with the HAND1 target genes, identified as above, and HAND1 ChIP-seq data. The model was drawn using BioTapestry.

### Immunoblotting

EBs were lysed in RIPA buffer (ThermoFisher Scientific™) supplemented with Halt™ Protease and Phosphatase Inhibitor Cocktail (ThermoFisher Scientific™), DNAseI and 5 mM EDTA. Protein lysate was quantified using the Pierce™ BCA Assay Kit (ThermoFisher Scientifc™). 10–20 µg of protein lysate was run through MiniPROTEAN TGX Stain-free gels 4–12% gels (Biorad). Protein was transferred to nitrocellulose membranes using the Trans-Blot Turbo Transfer System (Bio-rad) then blocked in blocking buffer (5% milk/0.1% Tween-20/PBS). Membranes were incubated overnight at 4 °C in blocking buffer with the appropriate antibodies (Appendix Table S3). Appropriate HRP-conjugated secondary antibodies were added for 1 h at room temperature in blocking buffer. Blots were developed using ECL™ Prime Western Blotting Detection Reagent (Amersham™) and imaged using ChemiDoc™ Imaging System (Bio-rad).

### Immunocytochemistry and microscopy

EBs/cells grown on coverslips were fixed with 10% neutral buffered formalin (Sigma-Aldrich) for 20/15 min, respectively, followed by 3X PBS washes. EBs were permeabilised with 0.5% Triton X-100/PBS for 1 h at room temperature, whereas cells on coverslips were permeabilised for 8 min. Cells were blocked with with 4% donkey serum/PBS for 1 h at room temperature. Antibodies (Appendix Table S3) were incubated overnight at 4 °C in blocking buffer. The next day, EBs were washed 3X with PBS-tween, followed by incubation with AlexaFluor secondary antibodies (Appendix Table S3) for 1 h at room temperature. EBs were stained with 1 µg/ml Hoechst for 15 min at room temperature, then washed with PBS/0.1% BSA. EBs were mounted onto coverslips using ProLong™ Gold Antifade Mountant (Invitrogen). Images were taken with an Evos™ M5000™ Imaging System (Thermo Fisher Scientific) or on a Leica SP8 confocal microscope.

### Chromatin immunoprecipitation (ChIP) and ChIP-seq

ChIP was performed largely as previously described (Sullivan and Santos, 2020). In brief, cells were differentiated as EBs as described above to D3. 5 µM of SB (or an equivalent volume of DMSO for vehicle-only controls) was added on D2-D3. EBs were dissociated into single cells with 1X TryPLE Select and fixed with 1% formaldehyde (Pierce) for 10 min at room temperature, then quenched with 0.125 M Glycine (Fisher Scientific™) for 5 min at room temperature. Cells were centrifuged and washed three times with cold PBS, then the pellets snap-frozen in a dry ice/ethanol bath and stored at −80 °C until required. Pellets were thawed on ice and re-suspended in Sonication/IP buffer (140 mM NaCl, 50 mM HEPES pH 7.5, 1 mM EDTA pH 8.0, 1%, 0.1% SDS, 0.1% sodium deoxycholate, 1X protease inhibitor). $3 \times 10^6$ cells in 150 µl per TPX 1.5 ml tube (Diagenode) was sonicated for 40 cycles (30 s on/30 s off on High Power) using a Bioruptor (Diagenode). 10 µg of anti-AM-Tag antibody (Active Motif) was incubated with 40 µl Protein G Dynabeads (Invitrogen) for 3 h at room temperature on a rotator. Sonicated chromatin was diluted with 1.5 ml of Chromatin Dilution Buffer (25 mM Tris pH 7.5, 5 mM EDTA, 1% Triton X-100, 0.1% SDS) plus protease inhibitors and centrifuged at $13,600 \times g$ for 30 min at 4 °C. The supernatant was transferred to

a protein low-bind tube (Eppendorf) and incubated with antibody/bead complex overnight on a rotator in the cold room. Wash steps and reverse cross-linking was performed as previously described (Sullivan and Santos, 2020).

Libraries were prepared using MicroPlex Library Preparation Kit v3 (Diagenode) according to the Manufacturer's instructions. Library quality was assessed using a 4200 TapeStation (Agilent Technologies) and DNA quantified using the Qubit (ThermoFisher Scientific™). The adaptor indices enabled the multiplexing of the libraries, which were pooled prior to loading on to the appropriate flow-cell. This was then paired-end sequenced (59 + 59 cycles, plus indices) on an Illumina NovaSeq6000 instrument. Finally, the output data was demultiplexed and BCL-to-Fastq conversion performed using Illumina's bcl2fastq software (v2.20.0.422).

### ChIP-seq analysis

Reads were trimmed using Trim Galore! to remove sequencing adaptors and poor-quality bases. Reads were aligned to the human genome (hg38) using Bowtie2 (v2.5.3). Peaks were called using MACS2 (v2.2.9.1) using default settings. Overlaps with ATAC-seq regions were identified using the R IRanges package (v.2.36). Peak annotation was performed using ChIPSeeker (v3.19).

### Transcription factor motif enrichment analysis

From peaksets obtained using MACS, we used Bedtools multiinter to combine the replicates of each sample and return the replicated peaks. For the day 7.5 samples, regions unique to each population defined only by SOX17-Tom and NKX2-5-GFP were used for the motif enrichment analysis. In the final analysis, to obtain comparable significance values, all peaksets were reduced to the same number of regions (4562) by random sampling. Multiple samples were tested to ensure results were representative.

Motif enrichment analysis was performed using Homer (v4.4) 'findMotifsGenome.pl'. Options '-mask' and '-size 200' and default background regions were used. A $p$-value of $<1e-6$ using the default binomial statistic was selected as significant.

### Statistical analysis

Statistical details of experiments can be found in the Figure legends, including $p$-values and numbers of samples analysed. Unless noted otherwise, adjusted $p$-values of below 0.05 were treated as significant. Plots were generated with ggplot2 (v3.5.1) in R or GraphPad Prism 9.

## Data availability

Dataset EV1 is available in Supplementary Information and for download at: https://doi.org/10.48420/26364184. Analyzed RNA-seq data (day 7.5) is browsable via an online application https://shiny.its.manchester.ac.uk/cardiac-gene-profiler/2. Analyzed single cell RNA-seq data (wild-type cells) is browsable via an online application https://shiny.its.manchester.ac.uk/cardiac-gene-profiler/3. Raw sequencing data has been deposited in the ArrayExpress database at EMBL-EBI under the following accession numbers: RNA-seq data (day 7.5): E-MTAB-14276. ATAC-seq data (day 7.5): E-MTAB-14277. RNA-seq data (day 3): E-MTAB-14278. ATAC-seq data (day 3): E-MTAB-14279. ChIP-seq data (day 3): E-MTAB-14280. Single cell RNA-seq data: E-MTAB-14285. Processed single-cell RNA-seq data is currently available for download at: https://doi.org/10.48420/26935324.

The source data of this paper are collected in the following database record: biostudies:S-SCDT-10_1038-S44318-025-00409-0.

## Peer review information

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

## Acknowledgements

This work was supported and funded by the UK Research and Innovation (UKRI) Future Leaders Fellowship (MR/T041668/1 to MJB); the British Heart Foundation (FS/PhD/23/29429 to MJB and NB); an Engineering and Physical Sciences Research Council (EPSRC) and Medical Research Council (MRC PhD studentship (EP/L014904/1 to NP); and the Biotechnology and Biological Sciences Research Council (BB/T007761 to NB and BB/X016684 to NB and MJB). We thank all the core facilities staff who supported this research. We thank David Elliot and Christine Mummery for providing the NKX2-5-GFP hESC reporter line. We thank Mudassar Iqbal for support with the SCENIC analysis. We thank Andy Sharrocks for reviewing the manuscript.

## Author contributions

**Adam T Lynch**: Data curation; Formal analysis; Writing—original draft; Writing—review and editing. **Naomi Phillips**: Data curation; Formal analysis; Writing—original draft; Writing—review and editing. **Megan Douglas**: Data curation; Formal analysis. **Marta Dorgnach**: Data curation; Formal analysis. **I-Hsuan Lin**: Data curation; Formal analysis. **Antony D Adamson**: Methodology. **Zoulfia Darieva**: Methodology. **Jessica Whittle**: Methodology. **Neil A Hanley**: Conceptualization; Supervision; Funding acquisition; Writing—original draft; Writing—review and editing. **Nicoletta Bobola**: Supervision; Funding acquisition; Writing—original draft; Writing—review and editing. **Matthew J Birket**: Conceptualization; Data curation; Formal analysis; Supervision; Funding acquisition; Investigation; Writing—original draft; Writing—review and editing.

Source data underlying figure panels in this paper may have individual authorship assigned. Where available, figure panel/source data authorship is

listed in the following database record: biostudies:S-SCDT-10_1038-S44318-025-00409-0.

## Disclosure and competing interests statement

The authors declare no competing interests.

# Expanded View Figures

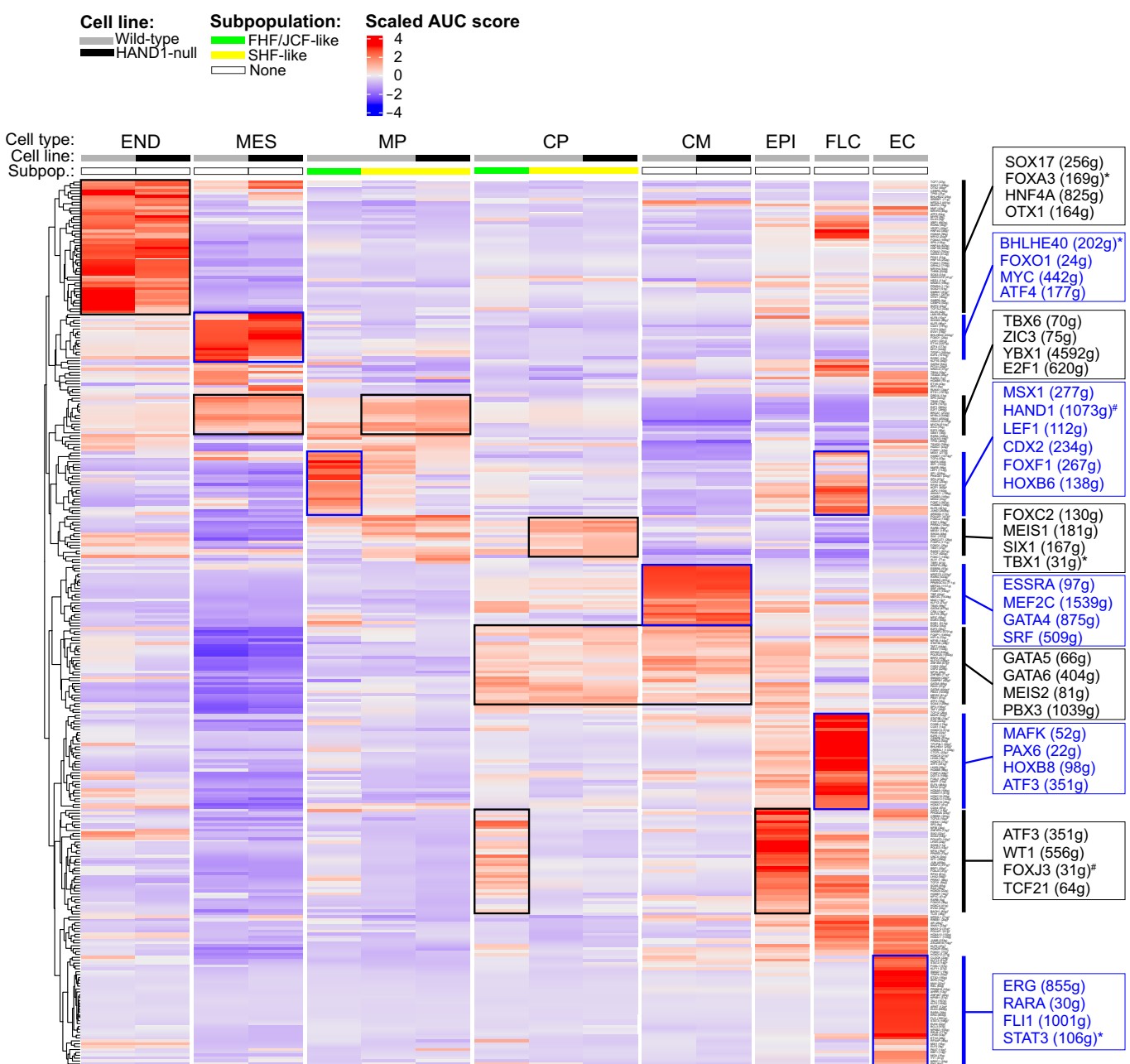

**Figure EV1. Regulatory network analysis during differentiation.**

Heatmap and hierarchical clustering showing the scaled target gene AUC score of the 307 regulons detected by SCENIC by cell line, cell type, and subpopulation. Clustering is by rows (regulons). AUC values were converted to a z-score and centred on the mean. Select markers present in clusters are highlighted. *Identified in HAND1-null population. #Identified in CPC lineage analysis. AUC area under the curve, CPC cardiac progenitor cell, CM cardiomyocyte, EC endothelial cells, END endoderm, EPI epicardial, FHF first heart field, FLC fibroblast-like cells, JCF juxta-cardiac field, MES mesoderm, MP mesodermal progenitors, SCENIC single-cell regulatory network interference and clustering, SHF second heart field.

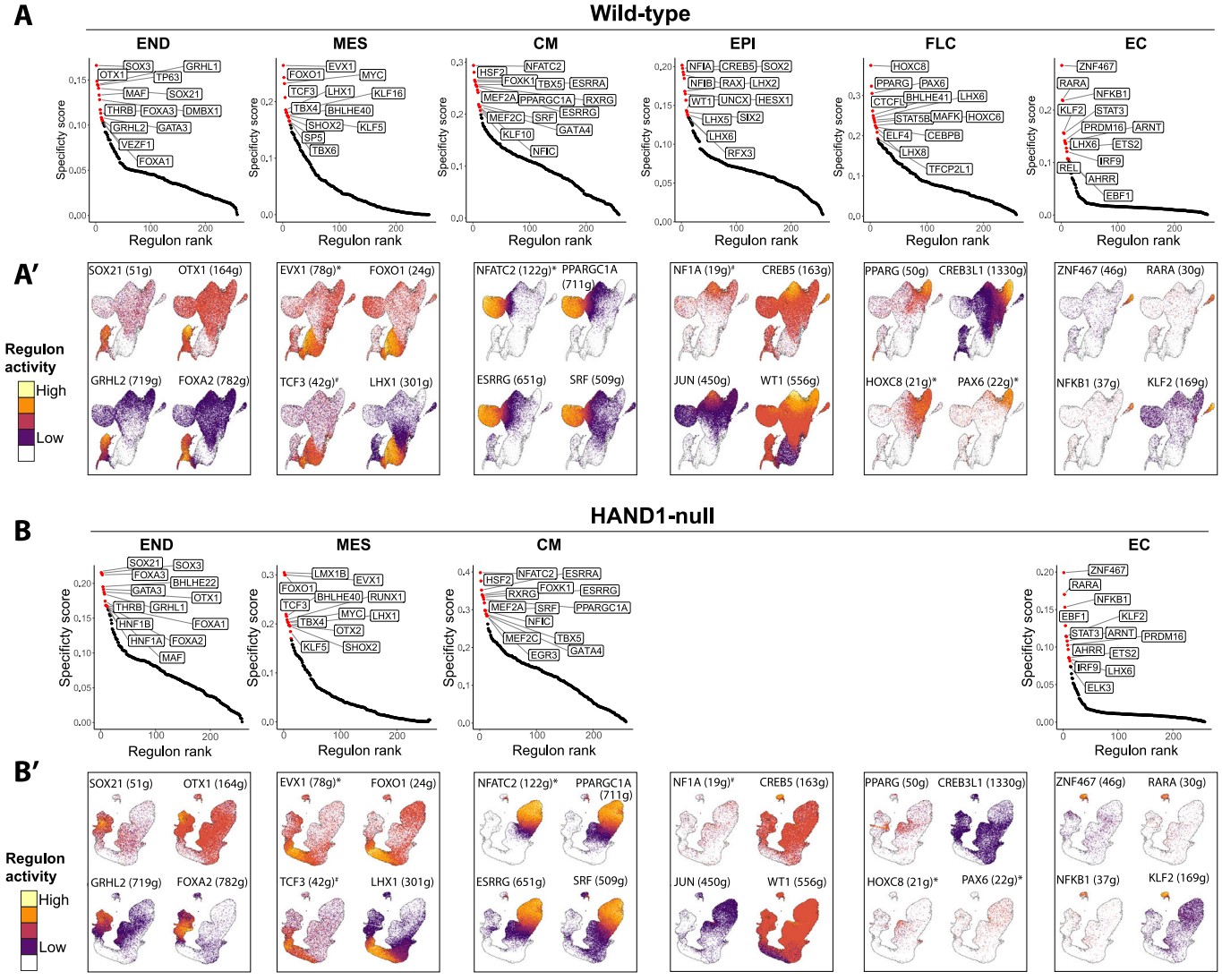

**Figure EV2.  SCENIC regulon specificity ranking in different populations during differentiation.**

(A) Regulon specificity scores ranked for different cell types in wild-type and (B) in HAND1-null populations. The HAND1-null line did not generate EPI or FLC types. The top 5% are highlighted red and several are labelled. (A') UMAP plots showing the activity of some key cell type-biased regulons in wild-type and in (B') HAND1-null cells. *Identified in HAND1-null population. #Identified in CPC lineage analysis. AUC area under the curve, CM cardiomyocyte, EC endothelial cells, END endoderm, EPI epicardial, MES mesoderm, FLC fibroblast-like cells, SCENIC single-cell regulatory network interference and clustering.

