## [Peer Review File · The EMBO Journal]

HAND1 level controls the specification of multipotent cardiac and extraembryonic progenitors from human pluripotent stem cells

Adam Lynch, Naomi Phillips, Megan Douglas, Marta Dorgnach, I-Hsuan Lin, Antony Adamson, Zoulfia Darieva, Jessica Whittle, Neil Hanley, Nicoletta Bobola, and Matthew Birket

Corresponding author: Matthew Birket (matthew.birket@manchester.ac.uk)

Review Timeline:

Submission Date:	4th Sep 24
Editorial Decision:	5th Nov 24
Revision Received:	13th Jan 25
Editorial Decision:	10th Feb 25
Revision Received:	12th Feb 25
Accepted:	20th Feb 25

Editor: Daniel Klimmeck

Transaction Report:

Dear Dr Birket,

Thank you again for the submission of your manuscript (EMBOJ-2024-118939-T) to The EMBO Journal. Please accept my apologies for getting back to you with unusual delay due to protracted referee input and detailed discussion in the editorial team. As mentioned earlier, your study was assessed by three reviewers with expertise in cardiac development and transcriptional control, whose comments are enclosed below.

As you will see from the experts' reports, the referees acknowledge the analysis and potential interest of your results. However, they also express major concerns regarding completeness of the findings, which need to be addressed thoroughly to make them supportive of publication in the EMBO Journal. The reviewers also raise issues related to the data presentation, additional controls and restructuring of the manuscript required, statistics applied and overall discussion of related literature, that would need to be conclusively addressed to achieve the level of robustness and clarity needed for The EMBO Journal.

Given the overall interest stated and broader angle of your findings, we are able to invite you to revise your manuscript experimentally to address the referees' comments. I need to stress though that we do require strong support from the referees on a revised version of the study in order to move on to publication of the work.

In light of the extensive experimentation requested, I would appreciate if you could contact me during the next weeks for exchange e.g. a video call to discuss your perspective on the comments and potential plan for revisions.

Please feel free to contact me if you have any questions or need further input on the referee comments.

When submitting your revised manuscript, please carefully review the instructions below.

Please feel free to approach me any time should you have additional questions related to this.

Thank you for the opportunity to consider your work for publication.

I look forward to your revision.

Best regards,

Daniel Klimmeck

Daniel Klimmeck, PhD
Senior Editor
The EMBO Journal

Instruction for the preparation of your revised manuscript:

- 1) a .docx formatted version of the manuscript text (including legends for main figures, EV figures and tables). Please make sure that the changes are highlighted to be clearly visible.
- 2) individual production quality figure files as .eps, .tif, .jpg (one file per figure).
- 3) a .docx formatted letter INCLUDING the reviewers' reports and your detailed point-by-point response to their comments. As part of the EMBO Press transparent editorial process, the point-by-point response is part of the Review Process File (RPF), which will be published alongside your paper.
- 4) a complete author checklist, which you can download from our author guidelines ([https://wol-prod-cdn.literatumonline.com/pb-assets/embo-site/Author Checklist%20-%20EMBO%20J-1561436015657.xlsx](https://wol-prod-cdn.literatumonline.com/pb-assets/embo-site/Author%20Checklist%20-%20EMBO%20J-1561436015657.xlsx)). Please insert information in the checklist that is also reflected in the manuscript. The completed author checklist will also be part of the RPF.

6) It is mandatory to include a 'Data Availability' section after the Materials and Methods. Before submitting your revision, primary datasets produced in this study need to be deposited in an appropriate public database, and the accession numbers and database listed under 'Data Availability'. Please remember to provide a reviewer password if the datasets are not yet public (see <https://www.embopress.org/page/journal/14602075/authorguide#datadeposition>).

7) Our journal encourages inclusion of *data citations in the reference list* to directly cite datasets that were re-used and obtained from public databases. Data citations in the article text are distinct from normal bibliographical citations and should directly link to the database records from which the data can be accessed. In the main text, data citations are formatted as follows: "Data ref: Smith et al, 2001" or "Data ref: NCBI Sequence Read Archive PRJNA342805, 2017". In the Reference list, data citations must be labeled with "[DATASET]". A data reference must provide the database name, accession number/identifiers and a resolvable link to the landing page from which the data can be accessed at the end of the reference. Further instructions are available at .

8) At EMBO Press we ask authors to provide source data for the main and EV figures. Our source data coordinator will contact you to discuss which figure panels we would need source data for and will also provide you with helpful tips on how to upload and organize the files.

Numerical data can be provided as individual .xls or .csv files (including a tab describing the data). For 'blots' or microscopy, uncropped images should be submitted (using a zip archive or a single pdf per main figure if multiple images need to be supplied for one panel). Additional information on source data and instruction on how to label the files are available at .

9) We replaced Supplementary Information with Expanded View (EV) Figures and Tables that are collapsible/expandable online (see examples in <https://www.embopress.org/doi/10.15252/embj.201695874>). A maximum of 5 EV Figures can be typeset. EV Figures should be cited as 'Figure EV1, Figure EV2" etc. in the text and their respective legends should be included in the main text after the legends of regular figures.

11) For data quantification: please specify the name of the statistical test used to generate error bars and P values, the number (n) of independent experiments (specify technical or biological replicates) underlying each data point and the test used to calculate p-values in each figure legend. The figure legends should contain a basic description of n, P and the test applied. Graphs must include a description of the bars and the error bars (s.d., s.e.m.).

We realize that it is difficult to revise to a specific deadline. In the interest of protecting the conceptual advance provided by the work, we recommend a revision within 3 months (3rd Feb 2025). Please discuss the revision progress ahead of this time with the editor if you require more time to complete the revisions.

Referee #1:

The authors have produced a vast amount of data to explore the role of different drivers in cardiac specification. They focus mostly on the role of HAND1 and made some discoveries that are promising. The problem is that the paper is too difficult to follow, with experiments not well explained and at some points the results are not convincing or are superficial. Again, in some ways the authors have explored many aspects of the biochemistry and cell biology of cardiac specification, but in my opinion it falls short of being rigorous on the conclusion. I have tried to point out many places where I think that improvements can be made. I suggest that the authors also streamline the story and make the experimental design clear, explaining why and which experiments are made to ask and answer questions.

The introduction is well organised and explains in general the background and aim of the paper. My few suggestions in this regard are:

- 1) Maybe the authors could elaborate a little bit more on the known role of HAND1 in lineage specification in general as well as other TFs that have been previously described to play a role in cardiac specification.
- 2) It will be good for the non-specialist in the field to introduce the signalling pathway of Activin A and BMP4 as they are used and inhibited from the onset.

Results:

The entire result section requires some work in my opinion. It is very difficult to follow in all aspects. It is not clear which experiments were done, which are the samples and why some experiments were even done.

I recommend that the authors in all the Result sections state how (which technique they used) to make clear to the reader what they are describing. For example we used WB for...or we titrated X by FACS....

Ideally the authors should stain the experiments of Figure 1d-g done in the dual cardiac-endoderm fluorescent reporter hESC line, with also some of the markers from figure 1b-c...in that way they can show that is not just the upregulation of MILX1 or FOXA2. This will strengthen the use of the line across all the following experiments.

The results in Figure 1h, require more characterization, maybe a few more markers?

Overall, I found the nomenclature confusing for the 8 cell populations, please describe them in the text clearly, rather than need to go to the figure and legend. Maybe they can be described better in the result section by using hematopoietic standard nomenclature (N+S+, N-S+ etc...). This becomes even more confusing in figure 2, did you only ATAC profiled 4 subpopulations? If this is the case, is not very clear in the text (actually from the text it seems that all populations were profiled, however in the figure it seems that only 4). I may have understood either wrongly, which indicates that is not clear.

It will be useful if a PCA for all the samples is provided in 1j is provided to understand how they relate to each other.

Please elaborate more on: We attribute the HAND motif enrichment to the expression of HAND1, and the HOX motif enrichment primarily to the expression of the HOXB cluster, principally HOXB1-6 (Extended Data Fig. 2b-d). Did you exclude others? Why?

It is not clear which are the experiments and how they answer the question. Furthermore, in figure 2b, how the identity of the differentiated was assessed?

The results presented in Figure 2e will need more characterization

In figure 3, it is clear that there are two populations missing in the HAND1 KO, however what is not clear is how do they know what those clusters are? In other words, how identity was assigned/annotated and verified. This is extremely important since one of the major conclusions is that: "there was a clear absence of epicardial and fibroblast-like cell development in the HAND1-null."

It is not clear why the UMAP has changed between figure 3 and 4. Please clarify.

Please can you add the scores/rankings of the SCENIC analysis. I

In figure 5 or the result section, it is not clear how many genes that change expression upon HAND1 expression seem to be the target of HAND1 (ChIP data with RNA-seq)? What happened to the rest of them?

How expandable are the HAND1-expressing progenitors? For how many passages they can still differentiate. How much do they change?

Discussion

As far as I could see, there is some literature already describing the effect of the concentration of ligands, signalling, pathways and TFs in lineage specification, such as the work by the groups of Keller, Barberi and Morris. I think that the results of this manuscript could be discussed in that context.

Across the result section and Discussion some statements are made without references.

Minor comments:

In the sentence: And what markers discriminate cardiac progenitors with different developmental potentials, including those with or without the ability to make epicardial cells?

It does not read well. Maybe it could be re-written as: And prompt us to ask: what markers discriminate cardiac progenitors with different developmental potentials, including those with or without the ability to make epicardial cells?

Line 125: Recognition motifs, is usually referred to as binding motif, I suggest to change it.

Line 142 Please spell out which lines in the text: To test this, we made a series of gene knockout hESC lines (Extended Data Fig. 3a-f).

Please explain this much better: To overcome the genetic redundancy of TEAD and HOX, we created YAP1-null hESCs to inhibit the TEAD-YAP1 complex, and generated a triple knockout of HOXB1-2-3 (HOXB1-3-null) to restrict HOX activity .

Referee #2:

- general summary and opinion about the principal significance of the study, its questions and findings

The study was designed to address the question of how different subsets of cardiac and non-cardiac lineage progenitors are specified in the human heart using hiPSC as a model, how they can be identified and how their functionality is determined. A population of multi-potent juxta-cardiac field progenitors was formed when the expression of the TF HAND1 was low that was able to make cardiomyocytes and epicardial cells whilst high HAND1 expression resulted in the formation of extraembryonic mesoderm. An interesting finding was that the HAND1 low population could be propagated in a multipotential state. The overall conclusion was that a TF-concentration dependent mechanism, in this case HAND1, determines cell fate. Identification of HAND 1 as a central regulator in this particular context is of interest and new even the aspects of the up- and downstream targets is somewhat expected. Overall, the work has been well carried out and the outcome interesting and relevant for human heart development. One shortcoming, is the statistical analysis and robustness of some experiments, as outlined below.

- specific major concerns essential to be addressed to support the conclusions

1. There are several concerns about the statistical analyses. Without being comprehensive, much of this needs checking. This includes but is not limited to for example: fig 1c, 1f and 1g are stated in the legends to be mean{plus minus}SD (n=2 biological repeats) but how is this possible? Fig 3: no stats mentioned.

Line 201 -202: Was the data in fig 4a and 4c significantly different? Using which stats?

2. All of the data and conclusions are based on a single cell line, which is understandable given the amount of work needed to generate all of the different knockout and reporter lines. However, robust conclusions on development based hiPSC lines generally requires that more than one be used to verify findings. The paper would be greatly improved by including experiments on the salient conclusions using another (independent) cell line. This might be knockdown rather than knockout for examples, specifically of HAND 1, to ensure the conclusions are correct.

- minor concerns that should be addressed

1. line 52 "hampering regenerative medicine": it is not only reg med that is affected but also disease modeling based on specific subtypes of heart cells affected by any particular condition

2. Regarding the expansion of the cardiac progenitor population by modulating MYC, it was not entirely clear whether this was only done in the hESC line expressing inducible c-MYC, or whether it could also be done using the regular hESC-NKX2.5-GFP line using the correct signaling pathway modifications. Please clarify more explicitly.

- any additional non-essential suggestions for improving the study (which will be at the author's/editor's discretion)

It is quite unusual these days to see hESC cultured on mitotically inactivated MEFs. Maybe the authors would care to comment. It can reduce mutation incidence but the reason may be simply historical.

Referee #3:

Lynch et al have used human pluripotent stem cells to study early fate choices in embryonic mesoderm and and identify HAND1 as a critical determinant of extraembryonic and epicardial versus cardiomyocyte fates. In a well documented study the authors reveal an interesting concentration-dependent role of HAND1, showing that high levels of HAND1 expression in their differentiation protocol promote extraembryonic and epicardial fates and low levels first heart field fates, while lack of HAND1 appears to promote second heart field fates. In addition, by manipulating MYC expression the authors succeeded in amplifying multipotent cardiomyocyte HAND1-low progenitor cells, of potential interest for cardiac repair strategies. This study illustrates the power of in vitro stem cell approaches to dissect early fate choices and makes a number of important findings that will be of interest to the general reader. The following points should be addressed.

1. Can the authors make any correlation between the levels of HAND1 expression driving extraembryonic versus cardiac fates in vitro and the situation in the embryo? This could be experimentally addressed in the mouse embryo where it might be anticipated that Hand1 expression increases along the medial to lateral embryonic axis and would add experimental validation to the authors' conclusions. At a minimum, embryonic patterning along the medial (BMP low) to lateral (BMP high) axis should be discussed.
2. Also concerning the mouse model, given the conclusion that the level of HAND1 is critical in determining fate, it would be useful to comment in the discussion on the lack of a Hand1 hemizygous mutant mouse phenotype. This appears to imply that 50% of Hand1 activity is enough to drive activate extraembryonic mesodermal and epicardial fates. Can the authors provide any quantification to show that this corresponds to the range of HAND1 levels they have implicated in the hPSC system? Can they make any conclusions as to whether the concentration-dependent effect is effective as a continuous gradient or demarcated by critical threshold concentrations of HAND1?
3. Please discuss whether there is any supporting evidence for expanded second heart field derived parts of the heart in homozygous Hand1 mutant mouse embryos?
4. Can the authors rule out the possibility that the SOX17 reporter is expressed in endothelial or endothelial progenitor cells?
5. Relevant to the observations on HOXB1-3, conditional misexpression of Hoxb1 in SHF-derived parts of the heart has been shown to modulate cardiac fate in reference 46. This should be commented on.
6. Please expand on the point about ectopic activation of HAND2 in a fraction of HAND1 null mesoderm cells (page 6). Can the authors distinguish between ectopic activation and amplification of another cell population? Please clarify to what extent this may reflect the switch to a SHF phenotype.

EMBOJ-2024-118939-T

Summary of new data added to the revised manuscript

- *HAND1* knockout made in an independent hESC line and differentiation characterised (**Appendix Fig. S3I-K**)
- Additional clone of *HAND1*-null HES3 hESCs also characterised (**Appendix Fig. S3H**)
- Additional experimental replicates performed for **Fig. 1C** (qPCR), **F** and **G** (FACS).
- SOX17 ICC performed and included in **Appendix Fig. S1A**.
- RT-qPCR of FHF and SHF markers in wild-type and *HAND1*-null progenitors (**Appendix Fig. S4B**)
- Interactive Shiny apps were made, and links included in the revised manuscript, for the day 7.5 FACS-RNA-seq data (**Fig. 1I, J**) and for the wild-type scRNA-seq timecourse (**Fig. 4A, B**) to enable user browsing and further data interrogation.
 - o <https://shiny.its.manchester.ac.uk/cardiac-gene-profiler/2>
 - o <https://shiny.its.manchester.ac.uk/cardiac-gene-profiler/3>

Minor data changes / corrections to note

- Error bars changed from SD to SEM in **Figures 1C, 1F, 1G, 2C** and **6I**.
- Concentrations of DMH1 corrected (10-fold lower than previously stated) and lowest concentration removed.
- Minor adjustments to the values in **Fig. 4A'** and **C'** after re-analysis.
- **Appendix Fig. S7B (previously Fig. S6B)** peak annotation values corrected.
- The *HAND1* regulatory region was adjusted from 0.5Mb to 0.1Mb and the *HAND1* regulon was slightly affected. All corresponding data was updated and ENSEMBL IDs were also added to the supplemental data table to facilitate future analysis.
- Due to a gene annotation issue we discovered when revising the manuscript (inconsistent IDs between platforms) some of the gene lists of *HAND1* targets have been changed by a small percentage. All changes are clearly marked.
- One of the FACS dotplots in **Fig. 6** (*HAND1*-low, CD31 vs *HAND1*-Tom) was replaced with a better example (as used in the quantification).
- Eight supplementary data figures (previously seven).

Referee #1:

The authors have produced a vast amount of data to explore the role of different drivers in cardiac specification. They focus mostly on the role of *HAND1* and made some discoveries that are promising.

We thank the reviewer for appreciating the scale and breadth of the work.

The problem is that the paper is too difficult to follow, with experiments not well explained and at some points the results are not convincing or are superficial. Again, in some ways the authors have explored many aspects of the biochemistry and cell biology of cardiac specification, but in my opinion it fall short of being rigorous on the conclusion.

We have carefully edited the results section to improve clarity and have added additional explanation where appropriate. We believe that the work is sufficiently rigorous and the conclusion that *HAND1* directs nascent mesoderm to juxta-cardiac field and extraembryonic lineages is well supported by several independent observations: *HAND1* binding motifs are enriched in the open chromatin of these cells; *HAND1* knockout causes the loss of these

lineages and of epicardial and fibroblast-like cells (confirmed by single cell RNA-seq); HAND1 directs a gene regulatory network including TFs enriched in the JCF and extraembryonic mesoderm.

Moreover, we can now show that *HAND1* knockout in an independent hESC line, which has been consistently intransigent to cardiomyocyte differentiation in 3D, was able to massively enrich cardiomyocyte differentiation in this format (**Appendix Fig. S3I-K**). This supports the universality of this mechanism.

We hope this level of rigor will come across more clearly in the revised manuscript.

I have tried to point out many places where I think that improvements can be made. I suggest that the authors also streamline the story and make the experimental design clear, explaining why and which experiments are made to ask and answer questions.

We thank the reviewer for highlighting areas where we could make improvements. We have carefully edited the results section to improve clarity and have added additional explanation where appropriate. We have paid particular attention to the first section of the results where we acknowledge the language needed improving. We have aimed to keep the writing concise to keep within a reasonable word length.

The introduction is well organised and explains in general the background and aim of the paper. My few suggestions in this regard are:

1) Maybe the authors could elaborate a little bit more on the known role of HAND1 in lineage specification in general as well as other TFs that have been previously described to play a role in cardiac specification.

We thank the reviewer for these suggestions. HAND1-null mice die due to a deficiency in extraembryonic mesoderm particularly in trophoblast development (Firulli et al., 1998; Riley et al., 1998) but its earlier role in mesoderm has not been previously explored. We have added a comment about this as well as about other TFs with more well-defined roles in cardiac specification.

2) It will be good for the non-specialist in the field to introduce the signalling pathway of Activin A and BMP4 as they are used and inhibited from the onset.

We have added an additional sentence and reference in the introduction to elaborate on this.

Results:

The entire result section requires some work in my opinion. It is very difficult to follow in all aspects. It is not clear which experiments were done, which are the samples and why some experiments were even done.

We accept that due to the large amounts of data within this paper, it may in some parts be difficult to follow. We have tried to introduce a rationale for each experiment and have made it clearer which techniques were used to address each question.

I recommend that the authors in all the Result sections state how (which technique they used) to make clear to the reader what they are describing. For example we used WB for...or we titrated X by FACS....

We have added some extra detail where appropriate to make it clearer what we are describing and the method of analysis. We hope that when combined with our detailed figure legends the descriptions are now sufficient.

Ideally the authors should stain the experiments of Figure 1d-g done in the dual cardiac-endoderm fluorescent reporter hESC line, with also some of the markers from figure 1b-c...in that way they can show that is not just the upregulation of MILX1 or FOXA2. This will strengthen the use of the line across all the following experiments.

We have added immunocytochemistry data of SOX17 co-stained with FOXA2 in EBs treated with SB and DMH1 (**Appendix Fig. S1A**). Co-expression is high as expected. Moreover, to make the FACS-RNA-seq data more accessible, we have created an online interactive app so that readers can access the entire dataset. We believe the data shows that the SOX17 reporter works exactly as expected.

<https://shiny.its.manchester.ac.uk/cardiac-gene-profiler/2>

The results in Figure 1h, require more characterization, maybe a few more markers?

We appreciate this suggestion and have tried to address this.

To compliment **Fig. 1H**, we have additionally stained for SOX17 (endoderm marker) in day 4 EBs as noted above (**Appendix Fig. S1A**) as well as cardiac troponin T (cardiomyocytes) in day 12 EBs (**Appendix Fig. S3G**). We additionally attempted to stain day 12 EBs with antibodies against fibroblast markers Decorin (DCN) and Lumican (LUM). However, staining was not successful (data not shown), which we attribute to technical unsuitability of these antibodies.

We have comprehensively analysed these populations by FACS-RNA-seq (**Fig.1E**). To be completely transparent this data (FACS-RNA-seq from day 7.5) is now easily accessible and browsable via our app as above. The scRNA-seq data also samples these exact populations. To make this equally accessible we have made another interactive app for browsing the wild-type scRNA-seq data:

<https://shiny.its.manchester.ac.uk/cardiac-gene-profiler/3>

Overall, I found the nomenclature confusing for the 8 cell populations, please describe them in the text clearly, rather the need to go to the figure and legend. Maybe they can be described better in the result section by using hematopoietic standard nomenclature (N+S+, N-S+ etc...).

We are happy to accept the reviewer's suggestion in naming the populations towards the end of the first results section by using and combining the abbreviations: NKX2-5-GFP (G) and SOX17-Tom (T).

This becomes even more confusing in figure 2, did you only ATAC profiled 4 subpopulations? If this is the case, is not very clear in the text (actually from the text it seems that all populations were profiled, however in the figure it seems that only 4). I may have understood either wrongly, which indicates that is not clear.

ATAC-seq was performed on all 8 populations but as described on lines 162-164, we collated regions which were unique by population classes based solely on GFP and Tom status (to simplify the analysis and increase robustness). We agree that this should be clearer. We have edited the text and have expanded **Appendix Fig. S2C** with a schematic indicating how the 4 population classes were defined. We thank the reviewer for highlighting this area of confusion.

It will be useful if a PCA for all the samples is provided in 1j is provided to understand how they relate to each other.

We have added PCA data of the RNA-seq to **Appendix Fig. S2A**.

Please elaborate more on: We attribute the HAND motif enrichment to the expression of HAND1, and the HOX motif enrichment primarily to the expression of the HOXB cluster, principally HOXB1-6 (Extended Data Fig. 2b-d). Did you exclude others? Why? It is not clear which are the experiments and how they answer the question.

We have elaborated on the logic:

*“We attribute the HAND motif enrichment to the expression of HAND1, and the HOX motif enrichment primarily to the expression of the HOXB cluster, principally HOXB1–6, those being the predominantly expressed family members in that population class (**Appendix Fig. S2D–F**)”.*

Rather than including the expression of every HOX gene in **Appendix Fig. S2D**, access to the gene expression Shiny app will allow readers to view all these data.

<https://shiny.its.manchester.ac.uk/cardiac-gene-profiler/2>

Furthermore, in figure 2b, how the identity of the differentiated was assessed?

The differentiation was principally assessed by NKX2-5-GFP expression quantified by flow cytometry, and for the YAP1-null also PDGFR- α expression. GFP was used as a surrogate for committed cardiac progenitors and cardiomyocytes (as in **Fig. 1J**).

The results presented in Figure 2e will need more characterization.

We performed comprehensive scRNA-seq on these populations across 7 timepoints of differentiation. As already stated above, we attempted to perform immunofluorescence with fibroblast markers DCN and LUM. However, we were unable to obtain informative staining. We did however also measure cTroponinT, and present this in **Appendix Fig. S3G**.

We have now also added qPCR analysis of three new replicate differentiations which shows a consistent fate change to the SHF lineage in HAND1-null EBs (**Appendix Fig. S4B**).

In figure 3, it is clear that there are two populations missing in the HAND1 KO, however what is not clear is how do they know what those clusters are? In other words, how identity was assigned/annotated and verified. This is extremely important since one of the major conclusions is that: "there was a clear absence of epicardial and fibroblast-like cell development in the HAND1- null."

We agree with the reviewer that this is a major point. The identity of the clusters is supported by the data in **Figure 3C**, where epicardial and fibroblast-like cell markers are shown to be strongly enriched (see highlighted figure below).

Since WT1 (epicardial marker) was previously found to be present in wild-type EBs and absent in the HAND1-null EBs (**Fig. 2E**), this supported the correct identity of the epicardial cluster. For the fibroblast-like cells, the marker signature defining the cluster is very strong, but we have been cautious to only label them as fibroblast-like cells, as although clearly mesenchymal they do not come from the epicardial cells (our lineage trajectory data), hence

are not obviously cardiac fibroblasts. **Appendix Fig. S5 and S6** additionally show the gene regulatory networks active in these clusters, which also support unique identities for these clusters and the cell type assignments that we have made.

It is not clear why the UMAP has changed between figure 3 and 4. Please clarify.

The UMAPs in **Fig. 3** were generated from a combined analysis of wild-type and HAND1-null cells (using batch correction to align them), whereas those in **Fig. 4** were generated separately and using only hypervariable genes implicated in heart development. We have added an additional sentence explaining the difference in the UMAP plots.

“To gain further biological insight into cardiac cell lineage development, we performed line-specific analyses using only hyper variable genes (HVGs) implicated in cardiac development (~1000 genes), while retaining the previous cell type annotations. Monocle 3 was used to make new UMAP plots for each line, and to assign pseudotime and lineage trajectory data from root mesoderm (Trapnell et al., 2014).”

Please can you add the scores/rankings of the SCENIC analysis.

The SCENIC regulon AUC scores for each cell are available via the downloadable loom file (the last link in the **data availability section**). These data are stored in “colAttrs/regulonAUC”. It is a matrix of 316 x 110,517 hence too big to present by other means.

In figure 5 or the result section, it is not clear how many genes that change expression upon HAND1 expression seem to be the target of HAND1 (ChIP data with RNA-seq)? What happened to the rest of them?

We thank the reviewer for highlighting this point. We have made the results clearer in two ways, firstly by adding more information to the **Supplemental Data 1** file, where putative targets can now be easily filtered by the presence/absence of HAND1 binding, and secondly by adding the percentage values to **Fig. 5E**.

Of note, following additional analysis post submission and feedback on our bioRxiv preprint, we have modified the capture window to +/-100kb from the gene TSS. This change makes little difference to the final numbers since we discovered some genes had been previously lost due to an annotation issue. The core network was unaffected as the problem was mostly with obscure genes.

How expandable are the HAND1- expressing progenitors? For how many passages they can still differentiate. How much do they change?

They can be passaged once and still maintain differentiation potential, but we know from our previous work (Birket et al., 2015) that these progenitors tend to lose their potency with time in culture. We have not formally assessed this with the latest conditions or strictly by HAND1 status, but we anticipate the results will be similar. This is an ongoing topic of investigation in our lab and requires a comprehensive study to understand why potency is lost and how this might be overcome. This question was considered beyond the scope of the current manuscript.

Discussion

As far as I could see, there is some literature already describing the effect of the concentration of ligands, signalling, pathways and TFs in lineage specification, such as the work by the groups of Keller, Barberi and Morris. I think that the results of this manuscript could be discussed in that context.

We thank the reviewer for the suggestion. We have cited some literature on the concentration dependency of TFs in development:

“A dosage dependency at some regulatory elements but independence at others, is consistent with the behaviour of other TFs functioning in development (Hannon et al., 2017; Kathiriya et al., 2021; Naqvi et al., 2023).”

We have added additional references to the work of Gordon Keller.

“In summary, this study provides a comprehensive map of human cardiovascular development and builds on work in the field that has described BMP signaling as a key determinant of mesoderm patterning and heart lineage development (Lee et al., 2017; Yang et al., 2022).”

Across the result section and Discussion some statements are made without references.

We have reviewed the text and where we are referring to published data, we have ensured the sentences are correctly referenced. We have added an additional 14 references to the manuscript.

Minor comments:

In the sentence: And what markers discriminate cardiac progenitors with different developmental potentials, including those with or without the ability to make epicardial cells? It does not read well. Maybe it could be re-written as: And prompt us to ask: what markers discriminate cardiac progenitors with different developmental potentials, including those with or without the ability to make epicardial cells?

We have made this recommended change. We thank the reviewer for the suggestion.

Line 125: Recognition motifs, is usually referred to as binding motif, I suggest to change it.

We have made this recommended change.

Line 142 Please spell out which lines in the text: To test this, we made a series of gene knockout hESC lines (Extended Data Fig. 3a-f).

We have made this recommended change.

Please explain this much better: To overcome the genetic redundancy of TEAD and HOX, we created YAP1-null hESCs to inhibit the TEAD-YAP1 complex, and generated a triple knockout of HOXB1-2-3 (HOXB1-3-null) to restrict HOX activity.

We have clarified this as follows:

“To overcome the genetic redundancy of TEAD and HOX, which have 4 and 39 family members respectively, we created YAP1-null hESCs to inhibit the formation of TEAD-YAP1 complexes, and generated a triple knockout of HOXB1-2-3 (HOXB1-3-null) to restrict HOX activity”

Referee #2:

- general summary and opinion about the principal significance of the study, its questions and findings

The study was designed to address the question of how different subsets of cardiac and non-cardiac lineage progenitors are specified in the human heart using hiPSC as a model, how they can be identified and how their functionality is determined. A population of multi-potent juxta-cardiac field progenitors was formed when the expression of the TF HAND1 was low that was able to make cardiomyocytes and epicardial cells whilst high HAND1 expression resulted in the formation of extraembryonic mesoderm. An interesting finding was that the HAND1 low population could be propagated in a multipotential state. The overall conclusion was that a TF-concentration dependent mechanism, in this case HAND1, determines cell fate. Identification of HAND 1 as a central regulator in this particular context is of interest and new even *[though]* the aspects of the up- and downstream targets is somewhat expected. Overall, the work has been well carried out and the outcome interesting and relevant for human heart development.

We thank the reviewer for highlighting the soundness, novelty and broad interest of the study.

One shortcoming, is the statistical analysis and robustness of some experiments, as outlined below.

- specific major concerns essential to be addressed to support the conclusions

1. There are several concerns about the statistical analyses. Without being comprehensive, much of this needs checking. This includes but is not limited to for example: fig 1c, 1f and 1g are stated in the legends to be mean{plus minus}SD (n=2 biological repeats) but how is this possible? Fig 3: no stats mentioned.

The reviewer is correct in that these data represent n=2 biological repeats. This was sufficient for our purposes and the conclusions drawn from the data were confirmed with the subsequent experiments. Nevertheless, we have now performed one additional experiment for **Figures 1C** and **1F** and two additional experiments for **Fig. 1G**. We have modified the error bars to SEM which we think is the more informative statistic for these measurements.

Line 201 -202: Was the data in fig 4a and 4c significantly different? Using which stats?

SHF marker genes *FGF10*, *ISL1* and *TBX1* were all significantly more highly expressed in the SHF vs FHF lineage of the wild-type (analysis in **Appendix Fig. S4A** and **A'**) using the "nbTestSH" function (FDR < 0.05) from the sSeq package (Yu et al., 2013). The percentage of positive cells of these markers is highlighted in **Fig. 4A'**. Concerning the corresponding values in the HAND1 KO, we felt it was sufficient to show that the percentage of positive cells for these markers was at least as high on the main trajectory. We didn't perform a statistical test on this comparison because the experiment is essentially n=1.

We have now performed Chi-squared tests to compare positive / negative marker expression in wild-type SHF vs HAND1-null, but all the differences are significant (even for 40% ISL1+ WT SHF vs 42% ISL1+ HAND1-null, where p=0.016). This is because the number of cells in each lineage is very large (>3000 in each) therefore internally even small differences are significant. In our opinion, the qualitative data here is more important and it isn't appropriate to add these test results for unreplicated data (which is normal for scRNA-seq data given the cost).

As noted at the top, on reanalysis of our final datasets to address this question we found the previously stated % values were not all precise / up to date. This only amounted to a ~2% max difference but we have updated the figure and text, nonetheless.

To address how consistent the lineage fate difference is between the WT and HAND1-null we have performed three more differentiations to examine the expression of key markers at day 5 by qPCR. This confirmed a significant decrease in FHF marker *TBX5* in HAND1-null and conversely an increase in *ISL1*, *JAG1* and *TBX1* (**Appendix Fig. S4B**). The increases in *ISL1* and *JAG1* were statistically significant. Overall, this provides strong evidence that the change in cell fate is reproducible and formally significant.

2. All of the data and conclusions are based on a single cell line, which is understandable given the amount of work needed to generate all of the different knockout and reporter lines. However, robust conclusions on development based hiPSC lines generally requires that more than one be used to verify findings. The paper would be greatly improved by including experiments on the salient conclusions using another (independent) cell line. This might be knockdown rather than knockout for examples, specifically of HAND 1, to ensure the conclusions are correct.

We agree with the reviewer that this is an important final step. We have addressed this point in two ways: 1) by analysing an additional HAND1-null clone of the HES3 line and 2) by generating and analysing a HAND1-null clone from an independent hESC line (UMANe002-A1; where we have recently introduced a 5' NKX2-5-GFP knock-in reporter (Douglas et al., 2024). In **Appendix Fig. S3H**, we show that in control conditions these additional HAND1-null clones make EBs with increased NKX2-5-GFP, whereas WT1+ cells are greatly diminished. Moreover, the poor cardiomyocyte differentiation of the UMANe002-A1 line with the EB system in similar conditions (consistent over several differentiations) was dramatically improved by HAND1 knockout, showing that HAND1 may be a major disruptor of cardiomyocyte differentiation in suboptimal conditions (**Appendix Fig. S3J-K**). The results strongly support the conclusion that the effect of HAND1 knockout is independent of clone and genetic background.

- minor concerns that should be addressed

1. line 52 "hampering regenerative medicine": it is not only reg med that is affected but also disease modeling based on specific subtypes of heart cells affected by any particular condition

We agree and thank the reviewer for the suggestion. We have edited this sentence as follows:

"...hampering effective regenerative medicine and disease modelling requiring the accurate programming of specific human cardiac cell types."

2. Regarding the expansion of the cardiac progenitor population by modulating MYC, it was not entirely clear whether this was only done in the hESC line expressing inducible c-MYC, or whether it could also be done using the regular hESC-NKX2.5-GFP line using the correct signaling pathway modifications. Please clarify more explicitly.

We thank the reviewer for highlighting this omission. The cells still required *MYC* transgene expression despite the signalling pathway modifications. We have added the following sentence to the results to clarify this:

“The dependence on MYC transgene expression remained.”

- any additional non-essential suggestions for improving the study (which will be at the author's/editor's discretion)

It is quite unusual these days to see hESC cultured on mitotically inactivated MEFs. Maybe the authors would care to comment. It can reduce mutation incidence but the reason may be simply historical.

Indeed, we culture HES3 hESCs this way for historical reasons – the embryoid body protocol was previously optimised using this method. We are not alone in this – feeder-dependent PSCs are still used predominantly by Gordon Keller's lab (Yang et al., 2022).

We find that small changes in maintenance methods can have a big impact on how cells will respond during differentiation. We are working towards updating the model to feeder-free conditions to make the protocols more universally transferable.

Referee #3:

Lynch et al have used human pluripotent stem cells to study early fate choices in embryonic mesoderm and identify HAND1 as a critical determinant of extraembryonic and epicardial versus cardiomyocyte fates. In a well documented study the authors reveal an interesting concentration-dependent role of HAND1, showing that high levels of HAND1 expression in their differentiation protocol promote extraembryonic and epicardial fates and low levels first heart field fates, while lack of HAND1 appears to promote second heart field fates. In addition, by manipulating MYC expression the authors succeeded in amplifying multipotent cardiomyocyte HAND1-low progenitor cells, of potential interest for cardiac repair strategies. This study illustrates the power of in vitro stem cell approaches to dissect early fate choices and makes a number of important findings that will be of interest to the general reader. The following points should be addressed.

We thank the reviewer for highlighting the important findings made and the broad general interest of the study.

1. Can the authors make any correlation between the levels of HAND1 expression driving extraembryonic versus cardiac fates in vitro and the situation in the embryo? This could be experimentally addressed in the mouse embryo where it might be anticipated that Hand1 expression increases along the medial to lateral embryonic axis and would add experimental validation to the authors' conclusions. At a minimum, embryonic patterning along the medial (BMP low) to lateral (BMP high) axis should be discussed.

We thank the reviewer for posing this interesting question.

Hand1 lineage tracing has shown reporter activation (and Mesp1 co-expression) in E6.75 mouse embryos at the extraembryonic/embryonic boundary with the confluence of extraembryonic and splanchnic mesoderm (Tyser et al., 2021; Zhang et al., 2021). As the reviewer notes, the early lateral plate mesoderm and extraembryonic mesoderm are zones of higher BMP signalling activity relative to more medial mesoderm. Zhang et al. traced Mesp1+ cells by scRNA-seq and showed that Hand1 and Bmp4 were co-expressed and limited to the juxta-cardiac field and extraembryonic lineages, and Hand1 was shown to be absent in more medial somitic mesoderm. The level of Hand1 mRNA also seems to follow

that of *Bmp4*, with both being at their highest in the extraembryonic lineages. Tyser et al. showed similar patterns of expression as does independent data from the Marionilab as shown below (<https://marionilab.cruk.cam.ac.uk/organogenesis/>).

Also, when Zhang et al. reduced the tamoxifen concentration in their Hand1-CreERT2 system to the lowest level of activity, most of the reporter positive clones were present solely in the extraembryonic tissue (yolk sac), which might support a higher *Hand1* expression level in this tissue.

Our data showing that the level of HAND1 expression continues to be regulated by the BMP signalling level, suggests that a gradient of HAND1 expression level would also be expected along the medial to lateral embryonic axis across the zone of HAND1 expressing cells.

We have added the following text to the discussion:

“The continued regulation of HAND1 expression by BMP signalling is consistent with the gradient of BMP levels, and Hand1 mRNA expression, in mesoderm along the mediolateral embryonic axis, extending into extraembryonic mesoderm (Hadas et al., 2024; Tonegawa et al., 1997; Tyser et al., 2021; Zhang et al., 2021).”

2. Also concerning the mouse model, given the conclusion that the level of HAND1 is critical in determining fate, it would be useful to comment in the discussion on the lack of a *Hand1* hemizygous mutant mouse phenotype. This appears to imply that 50% of *Hand1* activity is enough to drive activate extremes embryonic mesodermal and epicardial fates. Can the authors provide any quantification to show that this corresponds to the range of HAND1 levels they have implicated in the hPSC system? Can they make any conclusions as to whether the concentration-dependent effect is effective as a continuous gradient or demarcated by critical threshold concentrations of HAND1?

We thank the reviewer for these interesting questions.

Hand1 het mutants are indeed described to have no apparent phenotype and are fertile. However, a *Hand1* hypomorph mutant with (30-40%) expression is embryonic lethal, predicted to be caused by poor placental function (Firulli et al., 2010). A compensatory upregulation of *Hand2* was documented in the latter which may in part explain the resistance in the previous heterozygotes. These data are consistent with a sensitivity to HAND1 dosage in development.

Considering the concentration-dependent effect of HAND1 in our model and how a gradient of expression is translated to binary fate choices, we have some insight, but it is a major area of our current/future work requiring the use of more quantitative assays.

We think that HAND1 is both a direct driver of fate but also acts to influence the extracellular signalling environment, e.g. by upregulating BMP4, thereby also affecting differentiation non-cell autonomously. This type of positive feedback may contribute to the reaching of thresholds of lineage-determining TF expression. However, if we segregate mesoderm / early CPs by HAND1 mRNA level we only see a gradient of target gene and lineage marker expression (**examples in figure below**). Therefore, the nature of the threshold is not obvious. There can be major discrepancies between protein and mRNA abundance with TFs, therefore the mRNA level of *HAND1* may not be an accurate surrogate of the protein level.

What would be useful would be to segregate the earliest cells on the different lineages and examine their gene expression (*HAND1* and targets), but currently we don't have sufficient resolution in our data to do that with adequate confidence, particularly towards the extraembryonic lineage. We are planning more scRNA-seq focused just on these early stages so hopefully we will be able to answer this question in a future publication.

To elaborate a bit on the above points we have added the following text to the discussion:

“A dosage dependency at some regulatory elements but independence at others, is consistent with the behaviour of other TFs functioning in development (Hannon et al., 2017; Kathiriya et al., 2021; Naqvi et al., 2023). Hand1 heterozygote mutant mice had no apparent phenotype but mice with a Hand1 hypomorphic allele with 30-40% mRNA expression were embryonic lethal with poor placental function (Firulli et al., 1998, 2010; Riley et al., 1998). These data suggest that some buffering can take place but there is a sensitivity to Hand1 dosage in development, which should now be closely studied at the protein level and may have a role in developmental defects.”

Figure for reviewers. Single wild-type mesoderm and CP1 cells grouped by HAND1 mRNA level. The expression level of key lineage markers is shown for each cell population.

SHF = second heart field

FHF/JCF = first heart field / juxta-cardiac field

ExEm = extraembryonic mesoderm

3. Please discuss whether there is any supporting evidence for expanded second heart field derived parts of the heart in homozygous Hand1 mutant mouse embryos?

As the second heart field was not discovered until 2001, the early studies of Hand1-null mouse embryos which were published in 1998 did not specifically consider or assess this impact. Much of the later work has been performed with heart-specific mutants e.g. Nkx2-5-Cre or α -MHC-Cre (McFadden et al., 2005), and therefore not designed to assess the role of Hand1 in mesodermal progenitors.

4. Can the authors rule out the possibility that the SOX17 reporter is expressed in endothelial or endothelial progenitor cells?

We thank the reviewer for making this important point. Unfortunately, we did not consider endothelial SOX17 mRNA expression at the time of those experiments. Partly because we knew that endothelial cells were not commonly generated using this differentiation protocol (in the absence of exogenous VEGF). To retrospectively address this, we have performed a new experiment to measure the overlap at day 7.5. The results show that, although with DMH1 there are few endothelial cells (<2% of total), indeed about half of the endothelial cells do express SOX17-Tomato. These cells could therefore represent a small 'contaminant', but based on this measurement this would have only amounted to ~2% of the SOX17-Tom+ fraction. Reassuringly, in our sorted cell RNA-seq, endothelial gene expression (PECAM1, CDH5) was not enriched in the SOX17-Tom+ fractions.

Gene:

PECAM1

In hindsight, the sensible thing would have been to include a CD31-APC antibody labelling during these experiments to gate out those cells. However, based on the data above we

think the results would have been hardly affected. Our new publicly available app allows readers to freely interrogate the data:

<https://shiny.its.manchester.ac.uk/cardiac-gene-profiler/2>

5. Relevant to the observations on HOXB1-3, conditional misexpression of Hoxb1 in SHF-derived parts of the heart has been shown to modulate cardiac fate in reference 46. This should be commented on.

We thank the reviewer for the suggestion. We have added the following (in **bold**) to the discussion:

*“However, Hoxb1 is required to maintain proliferation in the posterior SHF in mice (Stefanovic et al., 2020) and was observed in the JCF (Tyser et al., 2021). The HOXB1–3-null showed that HOX TFs may play a significant role in restricting cardiac differentiation in a high BMP signaling environment. **This is also consistent with the effect observed when Hoxb1 is mis-expressed in the mouse anterior SHF or during in vitro differentiation (Darieva et al., 2024; Stefanovic et al., 2020).**”*

6. Please expand on the point about ectopic activation of HAND2 in a fraction of HAND1 null mesoderm cells (page 6). Can the authors distinguish between ectopic activation and amplification of another cell population? Please clarify to what extent this may reflect the switch to a SHF phenotype.

We thank the reviewer for making this point, which is one we hadn't previously considered.

Based on some markers, early *HAND2* expressing mesodermal progenitors (in the *HAND1*-null) appear more like WT *HAND1* expressing cells, therefore our impression was of a mechanism of compensation. An upregulation of *Hand2* mRNA in *Hand1*-null mice was also previously reported and considered compensatory (Firulli et al., 2010; Morikawa and Cserjesi, 2004). However, on investigating our data more thoroughly we conclude that we cannot adequately distinguish between ectopic activation and a switch to a SHF lineage, and therefore we should not make this claim.

We have removed the related sentence in the results and changed the discussion to say: *“despite some potential compensation by HAND2”*.

Dear Dr Birket,

Thank you for submitting your revised manuscript (EMBOJ-2024-118939R) to The EMBO Journal. Your amended study was sent back to the three referees for their scientific re-evaluation, and we have received detailed recommendations from two of them, which I enclose below. Please note that while referee #1 got delayed and has not yet provided his/her additional remarks, we have editorially assessed your response to the critique raised by this reviewer and found the issues to be addressed satisfactorily. As you will see, the other experts state that the work has been substantially enhanced by the revisions and they are now in favour of publication, pending minor revision.

Thus, we are pleased to inform you that your manuscript has been accepted in principle for publication in The EMBO Journal.

I will let you know in case we still receive referee #1's pending rereport during the next few days.

Please consider the remaining minor point by referee #3 carefully and adjust the manuscript text where appropriate.

Also, we now need you to take care of a number of issues related to formatting and data presentation as detailed below, which should be addressed at re-submission.

Please contact me at any time if you have additional questions related to below points.

Thank you for giving us the chance to consider your manuscript for The EMBO Journal. I look forward to your final revision.

Again, please contact me at any time if you need any help or have further questions.

Best regards,

Daniel Klimmeck

>> Please limit the keywords for your study to maximally five.

>> Author Contributions: Remove the author contributions information from the manuscript text. Note that CRediT has replaced the traditional author contributions section as of now because it offers a systematic machine-readable author contributions format that allows for more effective research assessment. and use the free text boxes beneath each contributing author's name to add specific details on the author's contribution.

More information is available in our guide to authors.
<https://www.embopress.org/page/journal/14602075/authorguide>

>> Correct order of manuscript sections: Abstract / Keywords / Introduction / Results / Discussion / Methods / Data Availability / Acknowledgements / Disclosure and competing interests statement // References / Figure legends / Tables and their legends / Expanded View Figure legends

>> Data availability section: please move to the end of 'Methods' before 'Acknowledgments'. Update the data links.

>> References: please adjust reference format to EMBO Journal format, 10 authors et al. . DOIs should only be used for preprints and datasets that have not been published yet.

>> Please update the bioRxiv reference for the Darieva et al. (2024), and update the citation since in the meantime published as regular article.

>> Tables S1 - S5 should be removed from the manuscript text and added to the appendix, as Appendix Table S1 etc. .

>>Appendix file with ToC: please remove the red font in the final version.

>> Figure callouts: Please ensure that the figures and panels Fig 4C, 4D are called out in sequential order.

>> Consider additional changes and comments from our production team as indicated below:

- Figure legends:

1. Please note that the exact p values are not provided in the legends of figures 2A, C.
2. Please indicate the statistical test used for data analysis in the legend of figure 2A.
3. Please note that the box plots need to be defined in terms of minima, maxima, centre, bounds of box and whiskers, and percentile in the legend of figure 11.

Referee #2:

The work reported remains an interesting study of how the TF HAND1 has a concentration-dependent, fate determining function in mesodermal progenitors and what its gene regulatory network is: HAND1 low directs differentiation towards multipotent juxta-cardiac field progenitors able to make cardiomyocytes and epicardial cells while HAND1 high promotes the development of extraembryonic mesoderm.

Most of my comments concerned the robustness of statistics and repetition of the findings in independent hPSC lines. This was carried out and in fact revealed that in one "difficult to differentiate line", modification of HAND1 greatly increased differentiation to CMs. This clearly confirmed the point being made. No other major or minor comments on this revised version

Referee #3:

The authors have addressed my previous concerns in the revised manuscript. This careful study provides new insights into how HAND1 controls cell fate decisions in a stem cell model of cardiac and extraembryonic fate acquisition. My only remaining comment is that the authors should highlight in the title that their results are acquired using in vitro human stem cells (for

example adding "in human pluripotent stem cells").

Referee #2:

The work reported remains an interesting study of how the TF HAND1 has a concentration-dependent, fate determining function in mesodermal progenitors and what its gene regulatory network is: HAND1 low directs differentiation towards multipotent juxta-cardiac field progenitors able to make cardiomyocytes and epicardial cells while HAND1 high promotes the development of extraembryonic mesoderm.

Most of my comments concerned the robustness of statistics and repetition of the findings in independent hPSC lines. This was carried out and in fact revealed that in one "difficult to differentiate line", modification of HAND1 greatly increased differentiation to CMs. This clearly confirmed the point being made. No other major or minor comments on this revised version

We thank Referee #2 for their support of this work and their appropriate request to validate the main conclusions in an independent cell line.

Referee #3:

The authors have addressed my previous concerns in the revised manuscript. This careful study provides new insights into how HAND1 controls cell fate decisions in a stem cell model of cardiac and extraembryonic fate acquisition. My only remaining comment is that the authors should highlight in the title that their results are acquired using in vitro human stem cells (for example adding "in human pluripotent stem cells").

We thank Referee #3 for their input and support of this work. We have changed the title to **"The level of HAND1 controls the specification of multipotent cardiac and extraembryonic progenitors from human pluripotent stem cells"**.

Dear Dr Birket,

Thank you for submitting the revised version of your manuscript. I have now evaluated your amended manuscript and concluded that the remaining minor concerns have been sufficiently addressed.

I am thus pleased to inform you that your manuscript has been accepted for publication in the EMBO Journal.

Related, I would like to ask for your consent to keeping the rebuttal figures included in this file.

On a different note, I would like to alert you that EMBO Press offers a format for a video-synopsis of work published with us, which essentially is a short, author-generated film explaining the core findings in hand drawings, and, as we believe, can be very useful to increase visibility of the work. Please see the following link for representative examples and their integration into the article web page:

<https://www.embopress.org/doi/full/10.15252/emj.2019103932>

Finally, we have noted that the submitted version of your article is also posted on the preprint platform bioRxiv. We would appreciate if you could alert bioRxiv on the acceptance of this manuscript at The EMBO Journal in order to allow for an update of the entry status. Thank you in advance!

If you have any questions, please do not hesitate to contact the Editorial Office.

Best regards,

Daniel Klimmeck

Daniel Klimmeck, PhD
Senior Editor
The EMBO Journal

EMBO
Postfach 1022-40
Meyerhofstrasse 1
D-69117 Heidelberg
contact@embojournal.org
